# Lidar estimates of birch pollen number, mass and related CCN concentrations

Maria Filioglou[1], Petri Tiitta[1,*], Xiaoxia Shang[1], Ari Leskinen[1], Pasi Ahola[2], Sanna Pätsi[2], Annika Saarto[2], Ville Vakkari[3,4], Uula Isopahkala[1], and Mika Komppula[1]

[1]Finnish Meteorological Institute, Atmospheric Research Centre of Easter Finland, Kuopio, Finland
[2]The Biodiversity Unit of the University of Turku, Turku, Finland
[3]Finnish Meteorological Institute, Helsinki, Finland
[4]Atmospheric Chemistry Research Group, Chemical Resource Beneficiation, North-West University, Potchefstroom, South Africa
[*]now at Envineer Oy, Microkatu 1, 70210, Kuopio, Finland

Correspondence: maria.filioglou@fmi.fi

**Abstract.** Accurate representation of microphysical properties of atmospheric aerosol particles – such as number, mass and cloud condensation nuclei (CCN) concentration – is key to constraining climate forcing estimations and improving weather and air quality forecasts. Lidars capable of vertically resolving aerosol optical properties have been increasingly utilized to study aerosol–cloud interactions, allowing for estimations of cloud–relevant microphysical properties. Recently, lidars have
been employed to identify and monitor pollen particles in the atmosphere, an understudied aerosol particle with health and possibly climate implications. Lidar remote sensing of pollen is an emerging research field, and in this study, we present for the first time retrievals of particle number, mass, CCN, giant CCN (GCCN) and ultra–giant CCN (UGCCN) concentration estimations of birch pollen derived from polarization lidar observations and specifically from a PollyXT lidar and a Vaisala CL61 ceilometer at 532 nm and 910 nm, respectively.

A pivotal role in these estimations is played by the conversion factors necessary to convert the optical measurements into microphysical properties. This set of conversion parameters for birch pollen is derived from in situ observations of major birch pollen events in Vehmasmäki station in Eastern Finland. The results show that under well–mixed conditions, surface measurements from in situ instrumentation can be correlated with lidar observations at higher altitudes to estimate the conversion factors. Better linear agreement to the in situ observations was found at the longer wavelength of 910 nm which is attributed
to a combination of lower overlap and higher sensitivity to bigger particles compared to observations at 532 nm. Then, the conversion factors are applied to ground–based lidar observations and compared against in situ measurements of aerosol and pollen particles. In turn, this demonstrates the potential of ground–based lidars such as a ceilometer network with polarization capacity to document large–scale birch pollen outbursts in detail and thus to provide valuable information for climate, cloud, and air quality modeling efforts, elucidating the role of pollen within the atmospheric system.

## 1 Introduction

Pollen from genus Betula, commonly known as birch pollen, represent the most allergenic tree pollen type in North, Central and Eastern Europe (D'Amato et al., 2007). Currently, 8–16 % of the general population is sensitive to birch pollen, where half of which manifests respiratory and other allergy–related symptoms (Biedermann et al., 2019). The cross–reactivity of birch allergens with certain food can also trigger the pollen food allergy syndrome (Geroldinger-Simic et al., 2011; Wang et al., 2023), impacting further the quality of life of the people sensitive to this pollen type. The allergic symptoms are concentration dependent (Pfaar et al., 2017) thus, accurate information of the pollen load in the atmosphere is essential. Predictability of the pollination time and concentration in the air is not only important due to the adverse health effects but also for agricultural applications of certain crop species (Galveias et al., 2024) and its relevance to weather and climate. To this direction, climate change is projected to intensify the sensitization of the population to pollen even further (Lake et al., 2018; Zhang and Steiner, 2022) and may also make this type of bioaerosol relevant in aerosol–cloud interactions. Recent laboratory and model studies show that intact pollen grains and sub–pollen particles (SPPs) are likely to contribute to cloud processes and suppress precipitation (Wozniak et al., 2018), acting both as cloud condensation nuclei (CCN), giant CCN (GCCN), ultra–giant CCN (UGCCN) (Pope, 2010; Steiner et al., 2015; Griffiths et al., 2012; Mikhailov et al., 2019; Prisle et al., 2019) and, ice nuclei (IN) (Diehl et al., 2001, 2002; Pummer et al., 2012; Hader et al., 2014; Dreischmeier et al., 2017; Gute et al., 2020; Burkart et al., 2021).

To date, more than 1000 stations worldwide (> 600 located in Europe, https://oteros.shinyapps.io/pollen_map/, last accessed: 15 September 2024) are monitoring pollen during the pollen season close to the ground–level (Buters et al., 2018), with the majority of them using the Hirst–type sampler developed at the middle of the 20th century (Hirst, 1952). Although this way of sampling pollen has been standardised and it currently serves as the only reference method for pollen monitoring, it suffers from several drawbacks. Most frequently, data are available retrospectively within a week or more from the time of collection limiting the breadth of application and, the temporal resolution varies from a couple of hours to a day. Furthermore, the analysis of the pollen samples is labor intensive. Because of this, dispersion models have been exploiting pollen specific phenological and meteorological relationships to forecast the concentration and distribution of pollen in the atmosphere (Sofiev et al., 2013). Huffman et al. (2020) and Buters et al. (2024) present an overview of recently developed instruments promising the autonomous and continuous identification of pollen in real– or near–real–time. These instruments utilize a broad range of measuring principles; from digital microscopy and holographic images to elastic scattering and fluorescence spectra. The advantage of using elastic scattering and fluorescence techniques (Veselovskii et al., 2021) as part of a remote sensor is that they can further provide systematic information on the vertical distribution of pollen in the atmosphere; information that is currently missing and it is essential for model verification and assimilation procedures.

Polarization lidars which are active remote sensors have been increasingly utilized to study pollen, as the non–spherical structure of some of the pollen types induce moderate to strong laser depolarization (Sassen, 2008; Noh et al., 2013; Sicard et al., 2016; Bohlmann et al., 2019). Past efforts have focused on the optical properties of different pollen types as well as their vertical distribution in the atmosphere (Bohlmann et al., 2019; Shang et al., 2020; Bohlmann et al., 2021; Shang et al., 2022). Although the linear particle depolarisation ratio (PDR) of pollen has been the focus of laboratory–based studies

(Cao et al., 2010; Cholleton et al., 2022a, b), the characteristic PDR of some pollen species in atmospheric conditions has been only recently determined (Filioglou et al., 2023). The knowledge of PDR for different aerosol types is of paramount importance in lidar–based aerosol classification algorithms (Nicolae et al., 2018) and methodologies estimating the aerosol microphysical properties from lidar observations. In particular, in POlarization LIdar PHOtometer Networking (POLIPHON) method (Ansmann et al., 2012; Mamouri and Ansmann, 2016, 2017) the contribution of spherical and non–spherical particles to the observed optical effect is determined utilizing aerosol type dependent PDRs (Tesche et al., 2009). Then, estimations of the number, mass, CCN and IN concentrations for an aerosol type are possible, if specific conversion factors are known. These conversion factors have been usually determined from Aerosol Robotic Network (AERONET) climatologies of optical and microphysical properties (Shinozuka et al., 2015; Mamouri and Ansmann, 2015, 2016; Ansmann et al., 2019; He et al., 2023). To this end, conversion factors for pollen particles have not been estimated. In addition, the 30 µm upper size limitation in aerosol particle diameter of the AERONET inversion products may not be representative of the larger pollen species.

In this article, we extend the applicability of lidars to estimate the number, mass, and CCN–related concentration of birch pollen. The microphysical properties of birch pollen were estimated using a synergy of lidar observations and in situ aerosol instrumentation. Specifically, the conversion factors needed and estimates of the aforementioned microphysical properties for birch pollen were determined at 532 nm and 910 nm wavelengths utilizing observations from a PollyXT and a Vaisala CL61 ceilometer. The lidar–derived microphysical estimates of birch pollen were compared against in situ pollen and aerosol observations and further aided utilizing the mixing layer heights retrieved from a HALO Photonics Streamline Pro Doppler lidar.

The paper is organized as follows. A summary of the site location, instrumentation and methods are given in Sect. 2, with focus on the determination of pollen conversion factors from in situ observations. The results are presented in Sect. 3. Section 3 includes also a case study showcasing the eligibility of the lidar–derived pollen microphysical estimates. Discussion and concluding remarks are given in Sect. 4 and Sect. 5, respectively.

## 2 Instrumentation and methods

### 2.1 Site description

Between 2016 and 2023, six measurement campaigns were carried out at Vehmasmäki station in Eastern Finland (62°44′ N, 27°33′ E; 190 m above sea level), focusing on pollen (Fig. 1). The rural station is surrounded by broad-leaved and coniferous trees and it is located 18 km away from the city center of Kuopio. The site is equipped with a multi–wavelength PollyXT lidar (Engelmann et al., 2016), a Vaisala CL61 ceilometer, a HALO Photonics Streamline Pro Doppler lidar, and various in situ instruments for aerosol characterization up to 10 µm aerosol particles as well as meteorological quantities from the station and a 318 m tall mast. In addition, a holographic imaging instrument (ICEMET) was installed on site in 2021 allowing to determine the shape and size distribution of 5–200 µm aerosol particles. The station is operated by the Finnish Meteorological Institute since Autumn 2012 (Hirsikko et al., 2014) and it is part of EARLINET (Shang et al., 2022) and POLLYNET (Baars et al., 2016). During the measurement campaigns which lasted from March to August each year, the pollen type and concentration were

determined from the samples collected with a Hirst–type volumetric air sampler (hereafter Burkard sampler). The dominant pollen types over the site are alder (*Alnus*), birch (*Betula*), pine (*Pinus*) and, spruce (*Picea*) in spring and early summer, with herbaceous species such as *Poaceae* and *Urticaceae* later in the summer (Fig. 2). Typically, the aerosol load is low over the measurement location and aerosol particles are located mostly within the first two kilometres with occasional intrusions of smoke and dust particles in the free troposphere (Baars et al., 2016; Bohlmann et al., 2019; Shang et al., 2020). Therefore, the site presents favorable conditions for characterizing pollen particles and investigating their role in various atmospheric processes.

## 2.2 The PollyXT lidar

PollyXT (Engelmann et al., 2016) is a 12-channel high–power lidar allowing the estimation of the particle backscatter coefficient ($\beta$) at 355, 532, and 1064 nm wavelengths and the volume depolarization ratio (VDR) and particle linear depolarization ratio (PDR) at 355 and 532 nm wavelengths. Additionally, extinction coefficients at 355 and 532 nm are available during nighttime utilizing the Raman technique (Ansmann et al., 1992). Water vapour mixing ratio profiles can be retrieved during dark hours using the 407 nm Raman-shifted wavelength (Filioglou et al., 2017). This PollyXT version features a second near-field telescope retaining full overlap at about 120 m. For the near-field, the elastic 355 nm, 532 nm and the equivalent Raman-shifted wavelengths are detected. Information above 400 m is considered at 532 nm since PDR observations are detected at the far–field only. Observations from 355 nm were omitted as the combination of high overlap region which is at 800 m and low birch PDR introduced high uncertainty in the retrievals limiting the availability of cases and robust conclusions. The vertical resolution amounts to 7.5 m and the temporal resolution is 30 s. A detail description of the operating principle as well as uncertainties expected for the optical properties can be found at Engelmann et al. (2016) and Baars et al. (2016). To retrieve the necessary $\beta$ and PDR profiles utilized in this work, the backward Klett inversion was performed (Klett, 1981) on 2 h temporally averaged profiles. A constant lidar ratio (LR) of 60 sr was used for the inversion (Bohlmann et al., 2019; Shang et al., 2020, 2022) while information below 400 m above ground level was omitted.

## 2.3 The Vaisala CL61 ceilometer

The Vaisala CL61 ceilometer is a 910.55 nm single channel pulsed laser diode elastic lidar transmitting linearly polarized light into the atmosphere. An alternating polarizing sheet filter enables the recording of the return light in the same channel at two different polarization states, termed co–polar and cross–polar. This setup allows for the determination of both the attenuated backscatter coefficient and VDR. Full overlap is reached at about 300 m above ground level and raw profiles are available at a temporal resolution of 5 s (for the attenuated backscatter coefficient) and 10 s (for the VDR). The range resolution is 4.8 m.

To retrieve the $\beta$ and PDR profiles, the forward Klett inversion was performed using a constant LR of 60 sr (Wiegner and Gasteiger, 2015). The calibration factor required for the forward inversion was determined following the stratocumulus cloud method (O'Connor et al., 2004). A 5–10 % uncertainty is anticipated with this method to the particle backscatter coefficient (Hopkin et al., 2019; Filioglou et al., 2023). In order to increase the signal–to–noise–ratio (SNR) and harmonize the lidar observations to the temporal resolution of the Burkard sampler (see Sect. 2.5), a 2 h temporal averaging was considered when

retrieving the aforementioned optical properties. Information below 200 m above ground level was omitted. We only considered observations during 2021 and 2022, since during the birch pollen period in May 2023, the instrument experienced condensation in the main window making the calibration challenging and therefore those data were omitted to ensure high quality retrievals.

## 2.4 The HALO Photonics StreamLine Pro Doppler lidar

A HALO Photonics Stream Line Pro scanning Doppler lidar (Pearson et al., 2009) was located at Vehmasmäki station during the campaigns. This pulsed Doppler lidar operates at 1565 nm and is capable of scanning within a 20° cone from vertical, i.e. elevation angles 70°–90°. The minimum usable range of the instrument is 90 m, as the lower range gates are affected by the outgoing pulse and the maximum range is 9.6 km above ground level. Range resolution of the lidar is 30 m. The Doppler lidar was configured to perform a velocity azimuth display (VAD) scan with 24 azimuthal angles at 75° elevation angle every 15 minutes. Between VAD scans the lidar operated in vertical stare mode, alternating between co– and cross–polar receiver mode.

Data from the Doppler lidar was post–processed according to Vakkari et al. (2019) and a SNR threshold of 0.001 was applied to the vertically–pointing measurements. Horizontal wind profiles were retrieved from the VAD scans following Browning and Wexler (1968). Turbulent kinetic energy (TKE) dissipation rate profiles were calculated using the method by O'Connor et al. (2010) from the co–polar vertical stare measurements with the horizontal wind profiles from VAD scans. Instrumental noise contribution was estimated from SNR profiles according to Pearson et al. (2009) and subtracted from the vertical wind variance time series before the TKE dissipation rate calculation. The Mixing Layer Height (MLH) was estimated from the TKE dissipation rate profiles using a threshold of $10^{-4}$ $m^2 \cdot s^{-3}$, similar to previous studies (e.g., Vakkari et al., 2015).

## 2.5 Burkard sampler: Hirst–type volumetric air sampler

Airborne pollen was collected at 4 m above ground level utilizing a Hirst–type volumetric air sampler manufactured by Burkard Manufacturing Co. Ltd in the UK (Hirst, 1952). The sampling tapes were cut in sections representing full days and analysed with light microscopy. In the analyses, the pollen grains were identified at genus or family level by comparing their characteristic shape and individual features with known pollen. The sample tapes were counted on a bi–hourly basis, taking four randomized samples per each strip representing a 2 h time period (Mäkinen, 1981). An uncertainty of 30 % is anticipated in the pollen concentration with this type of pollen sampling (Buters et al., 2012; Tormo-Molina et al., 2013; Adamov et al., 2021; Triviño et al., 2023).

In addition, 35 randomly selected images were extracted between 08 and 17 local time (LT) on the 12th of May 2021 and between 10th of May 2023 03 LT and 11th of May 2023 03 LT for every hour to investigate the particle size distribution of birch pollen. The images were acquired from the Burkard sampler using an optical microscope equipped with a digital microscope camera. The microscope glide table was first placed on a random crosswise position and then repositioned lengthwise always one hour ahead to obtain images representing every hour of the samples. Then, the Olympus cellSens Entry imaging software was used to mechanically measure the geometrical diameters of the particles identified as birch pollen grains in each image.

## 2.6 Aerosol in situ observations

### 2.6.1 Aerosol size distribution

We measured the aerosol size distribution in the size range from 10 nm to 200 μm with three different instruments: a NanoScan scanning mobility particle sizer (hereafter NS; Model 3910, TSI Incorporated, USA), an optical particle sizer (hearafter OPS; Model 3330, TSI Incorporated, USA), and a digital in–line holographic imaging instrument (hereafter ICEMENT; University of Oulu, Finland, Kaikkonen et al., 2020). Samples for the NS and OPS were taken 5 m above ground and delivered via a stainless–steel line to the instruments inside an air–conditioned container at a combined flow rate of 1.8 l min$^{-1}$ (0.8 l min$^{-1}$ for NS, 1.0 l min$^{-1}$ for OPS). The ICEMET, in turn, was located on the roof of the container, and it measured at the same height as the inlets for the NS and OPS (5 m above ground). The NS and OPS measured one size distribution every 1 min, whereas the ICEMET recorded one hologram per second.

The NS size distribution (mobility diameter of 10–420 nm in 13 size bins) and OPS size distribution (optical diameter of 0.3–10 μm in 16 size bins) were combined in a similar way as in our earlier works (Filioglou et al., 2023; Leskinen et al., 2020), again by neglecting the last two bins of the NS aerosol size distribution and the first bin of the OPS aerosol size distribution, by converting the OPS optical diameters to geometric mean volume equivalent diameters (Alas et al., 2019), and by using the long–term average of 1.46–0.009i of the complex refractive index at Vehmasmäki (Filioglou et al., 2023).

The main components of the ICEMET are a 660 nm wavelength laser diode, which acts as a point light source, and an image sensor with a resolution of 2048 × 2048 pieces of 3.45 μm pixels. The light source and image sensor are inside opposite disk–like housings behind protective windows. The disks are 10 cm in diameter and at the distance of 3 cm from each other. The sensing region between the disks is a truncated pyramid volume. When the coherent light from the light source scatters from the objects in the sensing volume which is about 0.4 cm$^{-3}$ and interferes with the other parts of the light field, a hologram, a complex diffraction pattern, is formed and recorded on the image sensor, and later processed digitally. In this work, the diffraction patterns were processed by using the ICEMET Server software (Molkoselkä, 2020) releases 1.6.0-1.14.0, depending on the year, giving the size, shape, and location of each particle in the sensing region. The ICEMET used in this work had a theoretical effective particle detection size limit of 5.3 μm, and was equipped with a tail wing that turned the instrument according to the prevailing wind direction so that there was an open path for the particles to enter the sensing region. It must be noted that the ICEMET was not available during the birch pollen season in May 2022, therefore ICEMET observations in this work comprise those from the years 2021 and 2023.

### 2.6.2 Black carbon observations

Black carbon (BC) concentration was measured with an Aethalometer (hereafter AE; Model AE–31, Aerosol Magee Scientific, Slovenia). The instrument collects sample on a quartz fiber filter, illuminates the sample with light sources at seven wavelengths (370–950 nm), records the light attenuation, and outputs the BC concentrations at the seven wavelengths with a selected time resolution (5 min in this work). The BC concentration hereafter refers to the output at 880 nm, which was corrected for filter loading and multiple scattering in a similar way as in Leskinen et al. (2020). In the correction, a long–term average multiple

scattering correction factor of 4.75 at 880 nm at Vehmasmäki was used (Filioglou et al., 2023). The BC observations were used to filter out the presence of smoke particles during the pollen measurement campaigns.

## 2.7 The Vaisala FD12P Weather Sensor

To aid the analysis, co–located observations from a Vaisala FD12P Weather Sensor were also considered. The instrument is capable of deriving the visibility, precipitation type, intensity and duration of precipitation at the measurement location. In the present study, the FD12P visibility and precipitation information were used to exclude cloudy times. In particular, cases where the visibility was less than 2 km or the precipitation flag was not 0, which is indication of cloud development, precipitation or fog, were omitted.

## 2.8 Particle mass concentration calculations

### 2.8.1 Particle mass concentration from in situ observations

Surface particle mass concentration estimations of birch pollen were calculated utilizing aerosol size distributions from ICEMET observations. The 2 h temporally averaged aerosol number concentrations (dN) were converted to volume concentrations (dV) using the mean diameter (d) of each size bin following $dV = dN(d)\frac{1}{6}\pi d^3$. The volume concentration of aerosol particles in the range between 12 μm and 35 μm was further summed and multiplied by the mass density ($\rho$) of birch which is assumed to be $\rho$ = 0.8 g cm$^{-3}$ (Gregory, 1961), yielding the coarse-mode birch mass concentration. This size range is indicative of birch pollen considering discrepancies within the birch pollen family (Stiebing et al., 2022; Raith and Swoboda, 2023; Theuerkauf et al., 2024).

Rather than employing a fixed birch pollen size to estimate mass concentration from the Burkard sampler, the mean volume diameter in the range between 12 μm and 35 μm from ICEMET was utilized.

### 2.8.2 Particle mass and number concentration from lidar observations

Lidar–derived number and mass concentration methods have emerged the past 15 years. In particular, in Ansmann et al. (2011) a synergy of lidar–photometer observations was developed to estimate the mass concentration of aerosol particles (m). The method requires that the mass density ($\rho$), the extinction–to–volume conversion factor ($c_v$) and the particle extinction coefficient $\alpha = \beta(\lambda) \cdot LR(\lambda)$ for a specific aerosol type at a certain wavelength to be known according to $m = \rho \cdot c_v(\lambda) \cdot \alpha(\lambda)$. Where $\lambda$ is the wavelength. The goal of this study is to provide the conversion factor that permits the estimation of birch number and mass concentration from lidar observations at 910 nm utilizing observations from a Vaisala CL61 ceilometer. To retrieve the number concentration, a similar procedure is followed according to $n = c_n(\lambda) \cdot \alpha(\lambda)$, where n is the number concentration and $c_n$ denotes the extinction–to–number conversion factor.

### 2.8.3 The extinction–to–volume(number) conversion factor, $c_v$ ($c_n$)

To this end, $c_v$ is estimated using the relationship between the vertically integrated (column) particle volume concentration from photometric observations via the AERONET inversion and the layer mean particle extinction coefficient, $\alpha$, from lidar observations (Ansmann et al., 2019). Related to the volume concentration, the AERONET algorithm considers particles with radii up to 15 μm (i.e., diameters up to 30 μm). This size limitation may introduce a significant bias in the volume size distribution of aerosols exhibiting that size, such that in fresh volcanic plumes, resulting to a more than 100 % of underestimation in the lidar–derived mass aerosol load (Ansmann et al., 2012). Since some pollen types are larger than 30 μm, the AERONET inversion method may not be representative for this aerosol type. To tackle the issue, the volume aerosol size distribution from ICEMET was utilized considering the size range from 12 μm to 35 μm. The extinction–to–number conversion factor, $c_n$ has a similar retrieval procedure where the number aerosol size distribution from ICEMET was utilized, instead of the volume one.

The second required parameter for the $c_v$ ($c_n$) calculation is the $\alpha$ for the specific aerosol type. The birch extinction coefficient, $\alpha_{birch}$, was derived by polarization lidar observations based on the backward (forward) Klett–Fernald inversion method for PollyXT (CL61) observations, respectively, and the birch component separation method from Tesche et al. (2009). We assumed a simple, externally mixed two component aerosol when using this separation technique. For the separation, the PDR of birch pollen from (Filioglou et al., 2023) was utilized as the non–spherical aerosol component. A PDR of 0.03 was used as the spherical aerosol component (Shang et al., 2020; Bohlmann et al., 2019). Portin et al. (2014) have explored the chemical composition of the aerosol population in the area and found that sulfate, nitrate, ammonium and organics are present at Kuopio, about 20 km from Vehmäsmäki station, where the inorganic to total ratio was about 42%. To account for the lidar overlap height limitation, the MLH from HALO Doppler observations was employed. Cases with a MLH top higher than 400 (200 m) were considered for PollyXT (CL61) observations, respectively, at any point during the 2 h temporal averaging, in which the share of birch pollen from the Burkard sampler was more than 90 % in the pollen mixture. Moreover, possible dust and smoke intrusions were excluded utilizing BC observations from AE instrument (BC < 0.1 μg m$^{-3}$) and modelled dust optical depth (DOD, DOD < 0.03) provided by the WMO Barcelona Dust Regional Center (https://dust.aemet.es/products/daily-dust-products, last accessed 15 September 2024). From the surface up to the MLH top, the variation of $\beta_{birch}$ may largely fluctuate depending on the amount, type, and distribution of birch pollen in the aerosol mixture. In order to reduce the uncertainty introduced by these factors and increase the comparability of the lidar observations to the surface observations, we used the mean $\beta_{birch}$ between 400 and 450 m (200 and 250 m) above ground level for observations from PollyXT (CL61), respectively, which multiplied by a LR of 60 sr (Bohlmann et al., 2019; Shang et al., 2020, 2022) was converted to $\alpha_{birch}$.

## 2.9 CCN–related concentrations

### 2.9.1 CCN, GCCN and UGCCN estimation from in situ aerosol observations

For the estimation of CCN ($n_{CCN,birch}$), GCCN ($n_{GCCN,birch}$) and UGCCN ($n_{UGCCN,birch}$) number concentration, the number aerosol size distribution in the particle size range of 130 nm to 35 μm, $n_{0.13-35\mu m}$, particles between 1 μm and 35 μm, $n_{1-35\mu m}$, and particles between 10 μm and 35 μm, $n_{10-35\mu m}$ were considered, utilizing observations from NS+OPS+ICEMET,

OPS+ICEMET and ICEMET, respectively. Note that all CCN–related estimations consider that birch pollen grains, sub–micron birch SPPs and other biological material co–exist in the bioaerosol mixture without being able to be distinguished from each other with the current instrumental setup. The 130 nm size limit was chosen as most birch SPPs below this size remain inactive at 0.18 % supersaturation (ss) (Mikhailov et al., 2021). At this supersaturation, Mikhailov et al. (2021) found the hygroscopicity of birch pollen particles, kappa–value, to be $k = 0.13 \pm 0.02$ and an estimation of the activated particles can be made according to

$$n_{CCN,birch} = n_{0.13-35\mu m} \cdot ss^k \tag{1}$$

Since birch pollen is low in concentration in the atmosphere compared to other aerosol particles which may more actively contribute to the $n_{CCN}$ and since with the current instrumental setup we cannot denote the existence of SPPs of birch pollen and other biological material (e.g., spores, fungi, algae etc.) in the aerosol mixture, an extra step was necessary. To counterbalance the contribution of other particles, the $n_{CCN,birch}$ was estimated by subtracting the average $n_{CCN}$ on site at times when there was no pollen indication in Burkard observations or dust/smoke intrusions. A mean $n_{CCN}$ of 27 cm$^{-3}$ was estimated for Vehmäsmäki station in 2021 and 2023 years. This number was then used during birch pollen times in order to determine the $n_{CCN,birch}$. For the $n_{GCCN,birch}$ and $n_{UGCCN,birch}$, a mean concentration of 0.13 cm$^{-3}$ and $4 \cdot 10^{-4}$ cm$^{-3}$ was estimated, respectively. Furthermore at these large particle sizes, we consider that all particles are the reservoir of potential GCCN and UGCCN.

### 2.9.2 CCN, GCCN and UGCCN estimation from lidar observations

A similar procedure to that of the number and mass concentration was followed for the CCN–related estimations from the lidar observations following (Mamouri and Ansmann, 2016) methodology. Specifically, using the modified equations for birch pollen, the number concentrations of CCN, GCCN and UGCCN can be estimated as follows:

$$n_{CCN,birch} = f_{ss,birch} \cdot n_{0.13-35\mu m,birch,dry} \tag{2}$$

$$n_{GCCN,birch} = f_{ss,birch} \cdot n_{1-35\mu m,birch,dry} \tag{3}$$

$$n_{UGCCN,birch} = f_{ss,birch} \cdot n_{10-35\mu m,birch,dry} \tag{4}$$

with an enhancement factor, $f_{ss,birch} = 1.0$ for ss = 0.18 % and the number concentration $n_{0.13-35\mu m,birch,dry}$ (considering particles with radius between 130 nm and 35 μm), $n_{1-35\mu m,birch,dry}$ (considering particles with radius between 1 μm and 35 μm), and $n_{10-35\mu m,birch,dry}$ (considering particles with radius between 10 μm and 35 μm), respectively. For the calculation of the $n_{0.13-35\mu m,birch,dry}$, $n_{1-35\mu m,birch,dry}$ and $n_{10-35\mu m,birch,dry}$, the following equations were used.

$$n_{0.13-35\mu m,birch,dry} = c_{0.13-35\mu m,birch} \cdot \alpha_{birch}^{x_{0.13-35\mu m,birch}} \tag{5}$$

$$n_{1-35\mu m,birch,dry} = c_{1-35\mu m,birch} \cdot \alpha_{birch} \tag{6}$$

$$n_{10-35\mu m,birch,dry} = c_{10-35\mu m,birch} \cdot \alpha_{birch} \tag{7}$$

For the conversion of $\alpha_{birch}$ into $n_{0.13-35\mu m,birch,dry}$, $n_{1-35\mu m,birch,dry}$ and $n_{10-35\mu m,birch,dry}$, the conversion parameters $c_{0.13-35\mu m,birch}$, $c_{1-35\mu m,birch}$ and $c_{10-35\mu m,birch}$, and exponent $x_{0.13-35\mu m,birch}$ needed to be determined. Equation (5) assumes a linear correlation of $\log n_{0.13-35\mu m,birch}$ with $\log \alpha_{birch}$. These parameters were determined for each wavelength using the aerosol size distributions from NS+OPS+ICEMET, OPS+ICEMET and ICEMET, respectively.

## 3  Results

### 3.1  In situ birch pollen observations

#### 3.1.1  Birch pollen size distribution

In order to get a better insight on the comparability of the Burkard sampler and ICEMET observations with regard to the number and mass concentration, the birch pollen particle size distribution needs to be considered. Figures 3a-b show two light microscopy extracted images of the sampling tapes from Burkard sampler during high and low birch pollen concentration on the 12th of May 2021 around 09 LT and 10th of May 2023 around 17 LT, respectively. These images represent a fraction of the birch size distribution at the given time and were randomly selected from the Burkard samples (see Sect. 2.5). Translating the images into birch pollen size distribution considering a 1 μm bin size, it is evident that birch pollen appears in a range of particle sizes (Fig. 3c-d). The analysis of 35 such samples concluded to a mean birch pollen size of 21.0 μm over the measurement site (Fig. 3e). The minimum and maximum birch pollen size concluded via this method was 15.6 μm and 25.7 μm, respectively, showcasing over 10 μm difference in the birch pollen size. Primarily, the size of birch pollen depends on the type of birch tree, therefore regional discrepancies may be anticipated within the birch family. Previously, birch pollen has been found in the range of 17.3–35 μm (Stiebing et al., 2022; Raith and Swoboda, 2023; Theuerkauf et al., 2024). Then, meteorological conditions such as the relative humidity and temperature may further affect the size and shape of atmospheric pollen.

Equivalent particle size distributions from ICEMET are shown in Figs. 3c-e. For comparability reasons, the particle size distributions from Burkard and ICEMET are normalized. Both Burkard and ICEMET present similar particle size distributions, suggesting that ICEMET instrument is able to observe aerosol particles in the size range of birch pollen. Please note that while Burkard samples present a snapshot of the birch size distribution at a given time, ICEMET provides continuous monitoring of the aerosol particle size distribution and in this case a $\pm 15$ min averaging around the Burkard samples was considered. Therefore marginal discrepancies are anticipated. Also, both instruments sense the geometrical particle diameter, and therefore their diameters are directly comparable. Primarily, the aerosol particle size distribution from ICEMET observations presents

multi aerosol modes in the 12–35 μm size range which may possibly be a more realistic real–time representation of the birch pollen size distribution (Fig. 3e). Dust and volcanic aerosol presence is excluded using the modelled DOD and the OMPS Sulfur Dioxide ($SO_2$) Planetary Boundary Layer (PBL) available at https://worldview.earthdata.nasa.gov/.

### 3.1.2  Number and mass concentration comparison

Since the Burkard sampler and ICEMET present fundamentally different operating principles, it is essential to perform a 
comparison with regard to the number and mass concentrations as well. Figure 4a shows the progress of birch pollen season from Burkard (red line) and ICEMET observations (blue lines) from 11th of May 2021 at 08 LT to 15th of May 2021 at 08 LT. Cloudy, foggy and precipitation cases were removed using observations from the FD12P sensor located on site. ICEMET performed well over the intense birch pollen times, sufficiently mirroring the progress of birch pollen season seen in Burkard observations. This is not the case for the absolute concentration of birch pollen. Exploring further the absolute concentration
discrepancy in the 2 h data, the closest ICEMET concentration to the Burkard one within the 2 h time window, is also presented (dashed blue line). Undoubtedly, the agreement is much better but to this end, a solid conclusion on this discrepancy is not possible. It is not possible because a calibration standard for airborne pollen has not yet been developed and the Burkard methodology is not foolproof. For example, the better agreement shown in Fig. 4a may result from methodological procedures in Burkard during the data analysis (e.g., the randomized four small areas as representative of the 2 h) and up–scaling of these
areas to reflect the 2 h pollen concentration assuming a constant multiplication factor (Mäkinen, 1981). This approach may bias the pollen concentration. Moreover, the discrepancy in absolute concentration may result from the different temporal resolution between the two techniques in conjunction to boundary layer changes and the in–homogeneous distribution of pollen within this 2 h time frame.

To put the timeseries into perspective, Figs. 4b-c present the overall agreement of birch concentration between the two instru-
ments considering observations during 2021 and 2023. For the conversion of the Burkard birch pollen number concentration to mass, the mean volume diameter (MVD) from ICEMET between 12 μm and 35 μm was considered. There is a systematic offset between Burkard and ICEMET estimated particle concentrations (assuming Burkard observations are correct). This sim-ilar feature has been recently reported when Burkard observations were compared against newly developed automatic systems (Maya-Manzano et al., 2023).

## 3.2  Lidar birch pollen observations

### 3.2.1  Birch pollen conversion factors

Figure 5 summarizes the birch conversion factors for the number and mass concentration estimation utilizing the number and volume size distribution from ICEMET observations and equivalent mean birch extinction coefficients from the lidars. Despite the multiyear pollen observation availability on site, the conversion factors for the number/mass concentrations from PollyXT 
are extracted from 2-year observations during 2021 and 2023 while equivalent factors from CL61 ceilometer consider 2021 observations due to the instrument availability of the sensors involved. For the CCN–related conversion factors, the datasets

used for CCN and GCCN are during 2021 for both lidars while UGCCN dataset for PollyXT includes both 2021 and 2023. For CL61, UGCCN conversion factor is extracted from the 2021 dataset. Cases with a MLH lower than 400 m for PollyXT and 200 m for CL61, within the 2 temporal averaging as well as dust and smoke intrusions are excluded utilizing HALO Doppler lidar, AE observations and modelled DOD. A total of thirty eight (twenty five cases) with a 2 h temporal averaging each are considered at 532 nm and 910 nm wavelengths, respectively, where the mean conversion factors are indicated by the regression lines and summarized in Table 1. The relationship between the birch extinction lidar coefficient exhibit a linear relationship with the number and volume particle size distribution from ICEMET observations for the particle range 12–35 μm. The highest birch concentration on site, is represented with the topmost point in all four panels. It was observed by Burkard instrument during the 12th of May 2021 at 8 UTC (7-9 UTC). For 532 nm wavelength this point deviates from the linearity, and it can be due to the transitioning of the boundary layer during the 2 h time frame along with the non–uniformity of the aerosol layer and the wavelength sensitivity to the aerosol particle size population. This is not valid at 910 nm due to the combination of the lower overlap at CL61 and the higher sensitivity of this wavelength to bigger particles. For the derivation of the number and mass conversion factors, we consider birch pollen in the above–mentioned particle size range. Nonetheless, we cannot exclude that smaller birch pollen, fragments of it or other biological material particles are present in the aerosol mixture and therefore may contribute, due to their size in relation to the lidar wavelength, in the aerosol mixture and therefore to the estimated extinction coefficient. This impacts mainly the $c_n$ concentration factor rather than $c_v$, since the contribution of aerosol particles between 2.5–12 μm in the volume is insignificant (see Sect. 4).

Regarding the $c_{0.13-35\mu m,birch}$, $c_{1-35\mu m,birch}$ and $c_{10-35\mu m,birch}$, and exponent $x_{0.13-35\mu m,birch}$ needed for the CCN, GCCN and UGCCN concentration estimations, respectively, Fig. 6 presents the relationship between the birch extinction coefficient at 532 nm and 910 nm and the aerosol number concentration from 130 nm to 35 μm, from 1 μm to 35 μm and from 10 μm to 35 μm utilizing observations from NS+OPS+ICEMET, OPS+ICEMET and ICEMET, respectively. A total of sixteen (nineteen) cases were considered at 532 nm (910 nm) and the corresponding conversion factors are summarized in Table 1. Similar to the number/mass conversion factor estimation, cases where the MLH was lower than 400 m (200 m) for 532 nm (910 nm) are not considered. Also, cases with smoke or dust contribution are also excluded. In addition, for all the CCN–related conversion factors the number concentration was considered after subtracting the average number concentration of aerosols in the same size range over the measurement location (see Sect. 2.9). This was necessary since direct measurements of birch pollen fragments and other biological material and distinction of them from the background aerosol population are not available.

### 3.2.2 Lidar estimates of number, mass and CCN–related profiles during a birch outbreak

The birch conversion factors in Table 1 were applied to 4–day lidar observations between 11th of May 2021 at 08 LT and 15th of May 2021 at 08 LT in which birch pollen peaked at Vehmasmäki station in Finland. Figure 7 summarizes the lidar–derived estimates of all microphysical properties and equivalent observations from in situ instrumentation (Figs. 7g-i).

During 12th of May 2021, the highest concentration for birch pollen was recorded by Burkard over Vehmasmäki station (64380 particles m$^{-3}$ at 10-12 LT) and the second highest concentration in the nearby Kuopio pollen monitoring site (62°8′ N,

27°63′ E; 98 m a.s.l.), which presents a continuous 43–year–long airborne pollen monitoring dataset. As a reference, birch pollen concentrations of less than 6000 particles m$^{-3}$, are observed on site 95 % during pollen season. The lidar estimations of birch pollen concentration reveal that the number of pollen particles is greatest near the ground, decreasing as one moves upwards during a non convective boundary layer. A mean (min–max) CCN concentration of 2500 (429–4741) cm$^{-3}$ was estimated from the lidar observations at 910 nm during the day at 200 m above ground level with a 0.20 (0.09–0.29) cm$^{-3}$ and 20.9 (9.2–30.9) $\cdot 10^{-3}$ cm$^{-3}$ for the GCCN and UGCCN, respectively. During the nighttime of 13th of May 2021 and midday of the next day, there is still a notable birch pollen load but at the same time BC concentration (0.13–0.22 $\mu$g· m$^{-3}$) and DOD (0.05 to 0.09) also rise, indicating a complex aerosol mixture. In turn, this complicates the decomposition of the lidar profiles since both dust and birch induce high PDR values. Nevertheless, Figs. 7g-i present the surface micropshysical properties from the in situ synergy and Burkard. Similar to Sect. 3.1, the lidars are capable to follow the progress of birch pollen season even though the lowest information is available at 200 m or higher above ground level. This is particularly valid during a convective boundary layer or when boundary layer processes during nighttime between the surface layer and the residual layer present minimal discrepancies. In fact, the smaller the wind speed difference between the surface and the elevated layer is the better the agreement between these two height levels. In turn this implies that, during unstable atmospheric conditions, higher discrepancies between the lidar– and in situ–estimated quantities are anticipated, due to the long temporal averaging of non–uniform aerosol layers together with the sensitivity of the specific wavelength to the aerosol particle size distribution.

## 4 Discussion

So far, the most common method for sampling pollen particles has been point measurements at ground level using the Hirst–type collection technique. However, new in situ and remote sensing methods for pollen monitoring begin to emerge. The lack of a reliable calibration standard and the limitations of the Hirst–type collection method as a reference method make challenging the assessment of the accuracy of other pollen monitoring instruments. Consequently, aligning in situ and lidar observations with Burkard data is not an optimal approach and could lead to propagation of errors. In this direction, Maya-Manzano et al. (2023) report an offset between all 9 novel automatic pollen observational instruments with the classic Hirst–type observations. This has been also found in this study when ICEMET observations were compared against the Hirst–type collector. Thus, a robust calibration standard needs to be developed to improve the reliability of airborne pollen monitoring.

The concentration of pollen is a critical parameter for aerosol models and health–related applications. Estimating pollen levels using lidar observations enhances the validation and assimilation efforts, while providing timely information to the public about potential peaks of pollen season. In this study, we have provided the means to estimate the number and mass concentration of birch pollen from lidar observations assuming that birch pollen particles reside in the 12–35 μm size range. For estimating the mass concentration, even if smaller coarse birch pollen particles or other biological material are present in the atmosphere (> 2.5 μm diameter), the uncertainty in the $c_v$ factor at 910 nm is in the order of 5 % with a re–estimated $c_v$ of 1.90 ± 0.10, which is within the uncertainty range of the 12–35 μm size range. Assuming an AERONET equivalent particle size range (1.2–30 μm diameter), $c_v$ of 1.84 ± 0.08 is obtained which presents a 6 % discrepancy from the 12–35 μm size

range. In comparison, when using the AERONET method (level 1.5), a $c_v$ of $1.24 \pm 0.06$ is estimated. This is not the case for the number concentration estimation in which the inclusion of aerosol particles above 2.5 μm leads to an order of magnitude higher $c_n$ factor than the 12–35 μm size range, with the relationship between volume concentration and extinction not to be linear anymore.

In this study, the particle extinction coefficient was estimated by multiplying the particle backsactter coefficient with a LR of 60 sr for both wavelengths. The value of the LR is a mean statistical value at 532 nm estimated from Raman observations for mixtures of birch and background aerosols with unknown relative contribution and little to no wavelength dependence between 355 nm and 532 nm wavelengths (Bohlmann et al., 2019; Shang et al., 2020). To this end, no LR has been reported for the 910 nm wavelength. To account for both the LR uncertainty in 532 nm due to the birch share in the aerosol mixture in previous studies and a possible wavelength dependence between 532 nm and 910 nm wavelengths, a sensitivity study due to selection of LRs to the conversion factors has been added (Tables A1 and A2). It is apparent that an inappropriate selection of the LR can significantly influence the conversion factors, consequently affecting the accuracy of derived microphysical properties.

This is the first time that lidar observations are used to estimate the number concentration of potential CCN, GCCN and UGCCN of birch pollen. The CCN parametrization was restricted to supersaturation below 0.2 % but smaller particles can be activated at higher supersaturation resulting to higher number concentration of potential CCN (Mikhailov et al., 2021). We have also assumed that the elevated CCN concentrations during the birch pollen period are caused by the presence of SPPs and other biological material but with the current instrumental setup we cannot confirm the presence of sub–micron birch pollen. Essentially, the capacity of aerosol particles to act as CCN depends on their size, chemical composition, hygroscopicity, morphology and the supersaturation at the cloud layer, which in turn is influenced by updraft velocities. Previously, laboratory and model–based studies have confirmed the CCN and GCCN activity of birch pollen (Pope, 2010; Griffiths et al., 2012; Steiner et al., 2015). In Wozniak et al. (2018), given high enough number concentration of pollen fragments, a 32 % suppression in precipitation in clean continental aerosol conditions was foreseen. Nevertheless, the presence of pollen fragments in the atmosphere is not monitored. Under high relative humidity, pollen particles rapture but there is not yet atmospheric observations to enumerate their frequency, concentration and size distribution of these fragments. Here, we presume the existence of SPPs by comparing two periods, one with birch pollen and another one without it, but other biological particles co–exist in the aerosol mixture. Nevertheless, the lidar and in situ estimated birch CCN–related concentrations during the peak birch pollen season indicates a potential source of CCNs for atmospheric cloud processes which is not currently being considered.

Although there are no atmospheric studies of birch pollen CCN–related concentration, there is a plethora of studies for other aerosol particles. In this context, CCN concentrations in a central European city ranged from $160\,\mathrm{cm^{-3}}$ to $3600\,\mathrm{cm^{-3}}$ with an average of $820\,\mathrm{cm^{-3}}$ (Burkart et al., 2011). Enhancement in CCN concentration was seen in the coastal southeast Florida when biomass burning aerosol particles were present in the atmosphere ($1408 \pm 976\,\mathrm{cm^{-3}}$ (at ss=0.2 %) (Edwards et al., 2021). In Boreal forest, measured concentrations in the order of $10^2$–$10^3\,\mathrm{cm^{-3}}$ were found at ss=0.2 % (Sihto et al., 2011) with elevated concentrations anticipated during a fire episode (Kommula et al., 2024).

Regarding GCCN and UGCCN, giant sea salt particles with radius larger than 5 μm are reported in concentrations of $10^{-4}$ – $10^{-2}\,\mathrm{cm^{-3}}$ in Feingold et al. (1999). Higher concentrations were reported when stronger winds prevail (Smith et al., 1989),

while Gonzalez et al. (2022) reports a concentration of sea salt particles above 1 μm to be in the order of $10^{-1}$ cm$^{-3}$. Similar
to a marine boundary layer in which the 25 μm giant sea salt particles are well mixed in the surface layer (Lewis and Schwartz, 2004), birch pollen can be similarly well–mixed influenced by turbulence and convection, yet under a stable boundary layer concentrations may be diminished due to gravitational settling. Nevertheless, modelling and field studies have shown that pollen has the capacity to travel long distances and remain aloft for days. The birch GCCN and UGCCN concentration estimated at 200 m above ground level in this paper is 0.20 (0.09–0.29) cm$^{-3}$ and 20.9 (9.2–30.9) $\cdot 10^{-3}$ cm$^{-3}$ during 12th of May 2021, respectively. On this day BC, dust and volcanic intrusions were marginal thus birch pollen can be cloud relevant (in the order of $\sim 10^{-3}$ cm$^{-3}$) also in atmospheric conditions and therefore be able to affect cloud precipitation efficiency (Cotton and Yuter, 2009). Although this case is exceptional, at other times cloud–relevant concentrations could be achieved by adding up other pollen species together.

## 5 Conclusions

We expanded the applicability of polarization lidars to assess the microphysical properties of birch pollen utilizing a synergy of aerosol size distributions from novel in situ instrumentation. In line with POLIPHON method, it permits the profiling of birch number and mass concentrations as well as estimates of CCN, GCCN and UGCCN concentrations from single–wavelength backscatter polarization lidar observations at 532 nm and 910 nm. The pivotal conversion factors required to convert the optical into microphysical properties in POLIPHON method derived from a synergy of NS, OPS and ICEMET in situ observations which provided aerosol size distributions from 10 nm to 200 μm. Typically, conversion factors are obtained using AERONET climatologies. However, AERONET inversion products account for aerosol particle sizes up to 30 μm in diameter. To accurately account for pollen, it is essential to include larger aerosol particle sizes. The novel approach can be used as an alternative method to derive the conversion factors of other large aerosol particles, for example, volcanic ash particles and larger pollen types.

By selecting cases with a well-mixed boundary layer, surface measurements from in situ instrumentation were correlated against lidar observations at higher altitudes to determine the conversion factors. Although a linear relationship was observed across both wavelengths, the best agreement was seen at 910 nm. This was attributed to the lower overlap region and the higher sensitivity of this longer wavelength in detecting large aerosol particles. We should note that birch pollen grains, SPPs and other biological material all co–exist in the bioaerosol mixture without being able to distinguish with the current instrumental setup their individual optical effect. Therefore, efforts should be put in characterizing this effect, if any. Moreover, for the derivation of the conversion factors a LR is presumed. To this end, the actual LR wavelength dependence of birch pollen is not known and in this study we have tackled the issue by estimating the conversion factors for a range of LRs. Also, it remains to be investigated in detail the conversion parameters of other pollen particles having their optical properties characterized first.

Then, the microphysical properties of birch pollen were investigated using observations from a PollyXT lidar and a Vaisala CL61 ceilometer with polarization capability at Vehmasmäki, a rural site in Eastern Finland. The novel pollen retrieval technique developed holds particular significance for ground–based lidar networks such that of a ceilometer and space–borne lidars featuring polarization capability permitting the characterization of pollen microphysical and optical properties. In this way,

point measurements at ground level providing limited information to forecasting models and, health–related applications, can be broadened both in space and time utilizing the lidar technique.

*Data availability.* Level-1 CL61 ceilometer and Halo Doppler data between 11th and 15th of May 2021 are available through CloudNet portal (Komppula and O'Connor, 2023). PollyXT observations and various optical products are available at https://polly.tropos.de/. Level-3 data are available upon request.

*Author contributions.* MF conceptualized the original paper, analysed the PollyXT and CL61 ceilometer and FD12P data and performed the analysis considering all data sources. AL and PT were responsible and provided the in situ ICEMET aerosol data. AL provided the AE, NS and OPS data. VV provided the HALO Doppler data. AS, PA and SP analysed the pollen samples and provided the pollen data. MF, AL, 480 PT, XS, UI and MK were responsible for the lidars and ensured good in situ operation. They have also prepared the pollen samples during the pollen campaigns. MF prepared the manuscript with contributions from all co-authors. All authors were involved in editing the paper, interpreting the results, and the discussion of the manuscript.

*Competing interests.* No competing interests are present.

*Acknowledgements.* The authors gratefully acknowledge the support of the Finnish Research Impact Foundation through the Tandem In- 485 dustry Academia (TIA) program. Dust data and/or images were provided by the WMO Barcelona Dust Regional Center and the partners of the Sand and Dust Storm Warning Advisory and Assessment System (SDS-WAS) for Northern Africa, the Middle East, and Europe. We acknowledge ACTRIS for providing the dataset used in this study; the dataset was produced by the Finnish Meteorological Institute and is available from https://cloudnet.fmi.fi/ (last access: 15 September 2024).

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

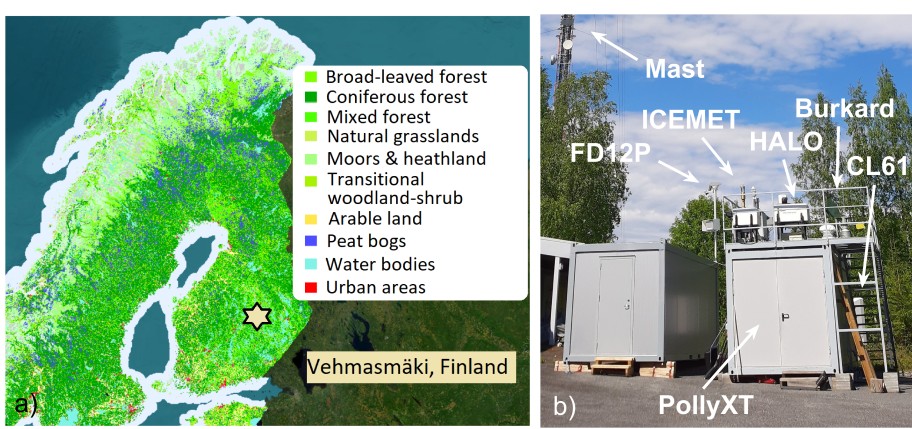

**Figure 1.** a) Measurement site location and land cover from the Copernicus Land Cover inventory CORINE in 2018 (https://www.eea.europa.eu/en/datahub/datahubitem-view/a5144888-ee2a-4e5d-a7b0-2bbf21656348, last access 10 November 2024). b) In situ and remote sensing instruments available on site. The Burkard sampler, ICEMET, Vaisala FD12P sensor and the HALO Photonics Doppler lidar are located on the roof of the main container at 4 m above ground level. The Vaisala CL61 ceilometer is located at the back of the main container. The PollyXT, NanoScan, OPS and AE instruments are located inside the container. In close proximity a 318 m tall mast equipped with weather sensors at different height levels provides profiles of various meteorological quantities.

.

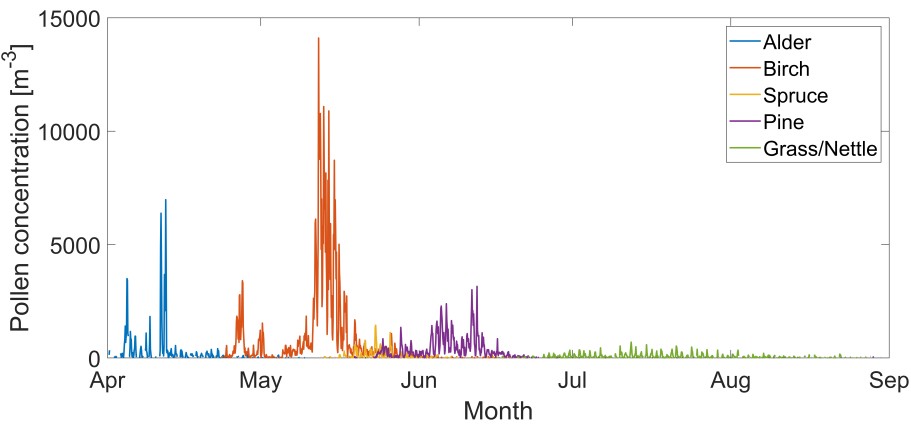

**Figure 2.** Timeseries of mean number concentration of the most common pollen species in Vehmasmäki, Finland utilizing six years of surface pollen observations from Burkard sampler.

.

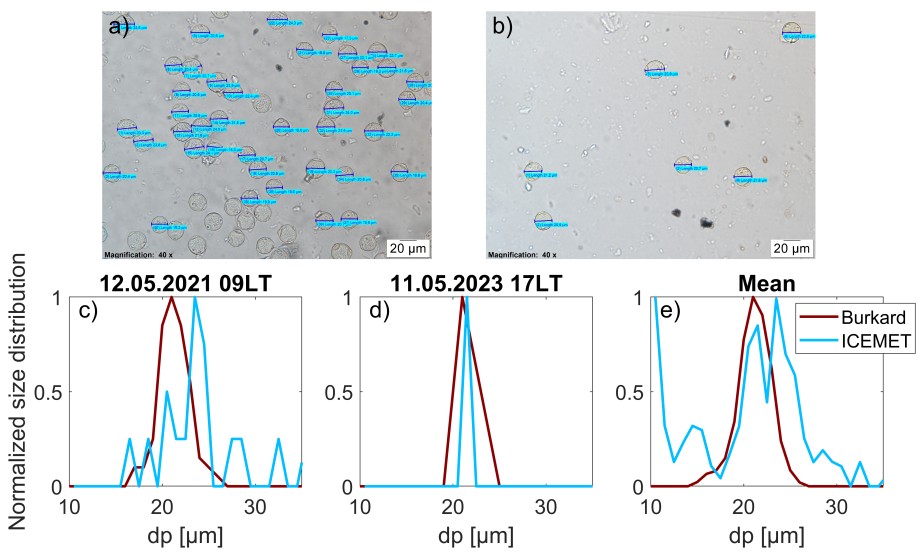

**Figure 3.** Microscope-extracted images of birch pollen from the sampling tapes collected with Burkard sampler on the a) 12th of May 2021 at 09 local time (LT) and b) 10th of May 2023 at 17 LT. The Olympus cellSens Entry imaging software was used to mechanically measure the geometrical diameters of the particles (blue lines) identified as birch pollen grains for up to 40 individual grains per sample. c-d) Burkard-estimated normalized birch pollen size distributions during the two aforementioned cases considering a particle size bin of 1 μm (solid red lines) and equivalent aerosol size distribution from ICEMET between 12 and 35 μm (solid blue lines). e) A 35–sample mean birch pollen size distribution from Burkard sampler. The samples where randomly extracted between 12th of May 2021 from 08 to 17 LT and 10th of May 2023 03 LT to 11th of May 2023 03 LT for every hour. Equivalent aerosol size distributions from ICEMET instrument for the same cases are marked with solid blue lines.

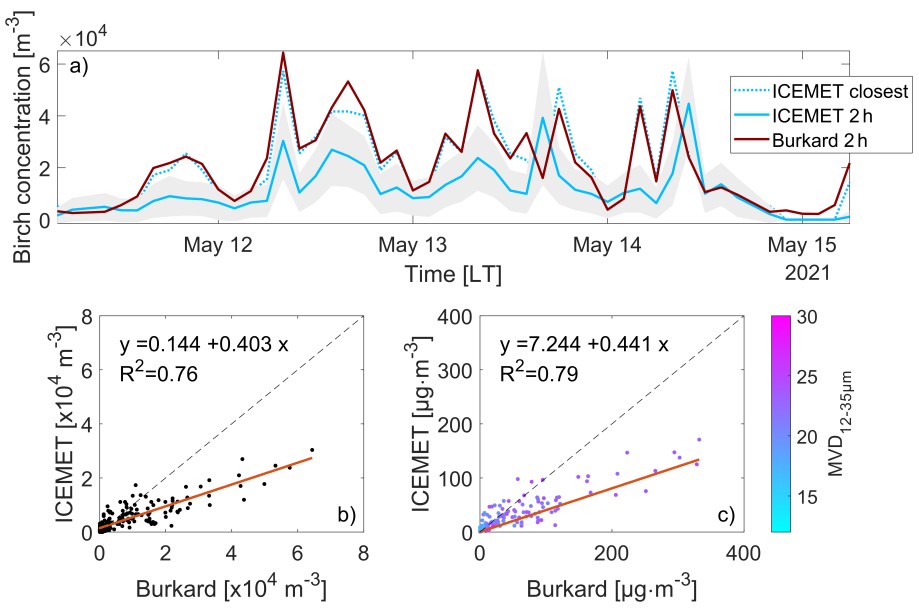

**Figure 4.** a) The progress of birch pollen season between 11th of May 2021 at 05 local time (LT) and 15th of May 2021 at 05 LT at Vehmasmäki station in Finland. The 2 h temporal progress of birch pollen season from Burkard is noted with the red solid line. The total ICEMET concentration between 12 μm and 35 μm size range at 2 h temporal resolution and the closest ICEMET concentration to that of Burkard within the 2 h time frame are marked with blue solid and dashed lines, respectively. The shaded area denotes the standard deviation within the two hours. b) Scatter plots between Burkard and ICEMET number concentration considering birch cases during 2021 and 2023 with a 2 h temporal resolution. c) Similar results for the mass concentration agreement between the two in situ instruments. To convert the number to mass concentration for Burkard observations, the mean volume diameter (MVD) between 12 μm and 35 μm size range from ICEMET was considered. Both b) and c) plots contain additional information regarding the fit and correlation of the two datasets. The dashed black line represents the 1:1 reference line

.

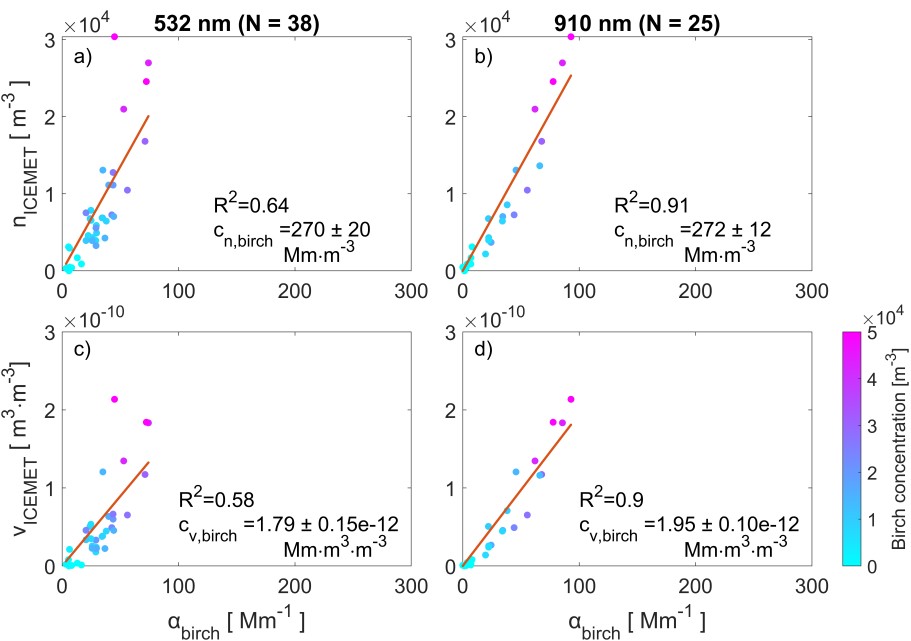

**Figure 5.** Relationship between birch extinction coefficient $\alpha_{birch}$ at 532 nm and a) particle number concentration $n_{ICEMET}$, c) volume concentration, $v_{ICEMET}$ considering particles between 12 and 35 μm. Correlations are shown utilizing the mean birch extinction coefficient between 400–450 m above ground level. The slope indicates the conversion factors $c_{n,birch}$ and $c_{v,birch}$ and are also given as numbers in the panels along with the goodness of the fit expressed through $R^2$ statistical value. Equivalent results at 910 nm are given in panels b) and d).

.

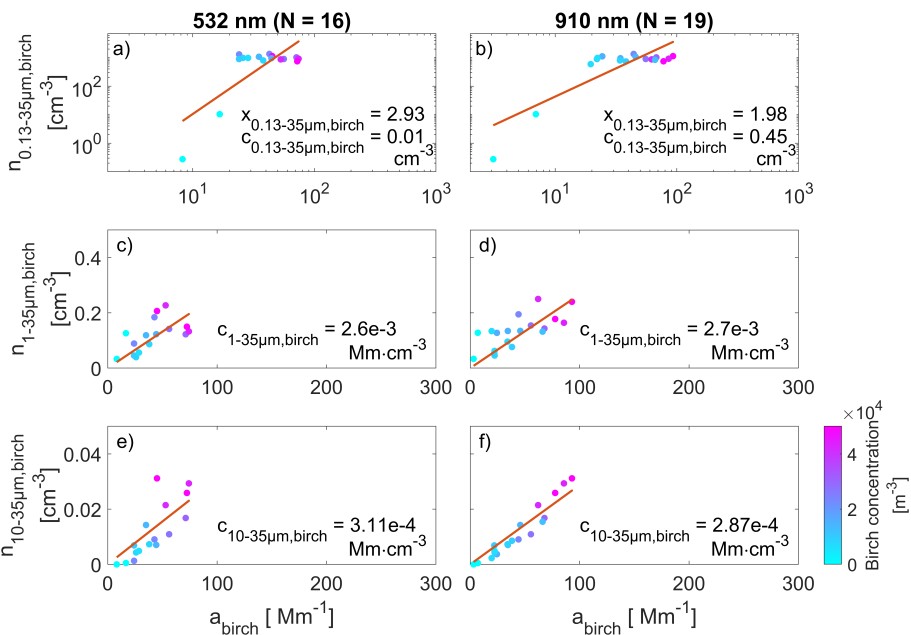

**Figure 6.** Relationship between birch extinction coefficient $\alpha_{birch}$ at 532 nm and particle number concentration between a) 0.13 μm and 35 μm. c) 1 μm and 35 μm, e) 10 μm and 35 μm. Correlations are shown utilizing the mean birch extinction coefficient between 400–450 m above ground level. Equivalent results at 910 nm between 200–250 m for the birch extinction coefficient are given in panels b), d) and f). In panels a) and b) the regression analysis is applied to the log(n)–log($\alpha_{birch}$) data. The conversion factors $c_{0.13-35\mu m,birch}$, $c_{1-35\mu m,birch}$, $c_{10-35\mu m,birch}$ and $x_{0.13-35\mu m,birch}$ indicate the intercept and slope of the regression, respectively, for each size range and they are also summarized as numbers in each panel.

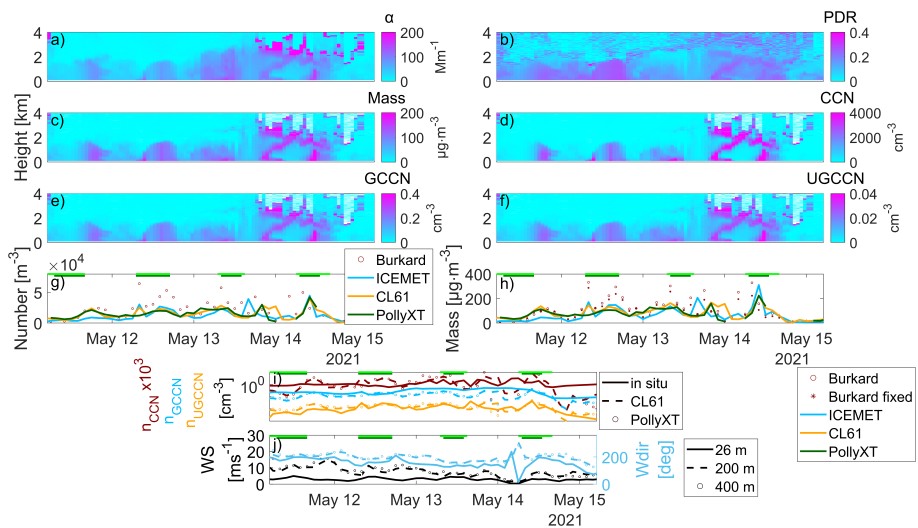

**Figure 7.** Timeseries of optical and estimated microphysical aerosol properties between 11th of May 2021 at 05 LT and 15th of May 2021 at 05 LT at Vehmasmäki station in Finland. a) Total particle extinction coefficient at 910 nm from CL61 ceilometer observations. b) PDR at 910 nm. c) Mass concentration estimated from 910 nm. d) CCN concentration estimated from 910 nm. e) UCCN concentration estimated from 910 nm. f) UGCCN concentration estimated from 910 nm. g) Comparison between in situ (Burkard and ICEMET) and lidar estimated number concentrations. h) Comparison between in situ (Burkard and ICEMET) mass concentrations at the surface and equivalent lidar estimated mass concentrations. For the Burkard mass concentration estimation both the MVD from ICEMET (circles) and a fixed birch pollen size of 22 μm (stars) is considered. For the CL61 ceilometer observations, a mean mass concentration between 200 and 250 m above ground level is considered. For the PollyXT lidar observations, a mean mass concentration between 400 and 450 m above ground level is considered. i) CCN, GCCN and UGCCN estimations from NS+OPS+ICEMET, OPS+ICEMET and ICEMET at the surface and equivalent estimations from the PollyXT (CL61) at 400–450 m (200–250 m) above ground level are shown, respectively. j) Wind speed and direction at the surface (26 m) from the mast observations and at 200 m and 400 m from HALO Doppler lidar. The CL61 ceilometer data shown in a–f are retrieved with an 1 h temporal resolution while plots g-j with a 2 h temporal resolution. The times that the mixing layer height was above the 400 m (200 m) height level are indicated by dark green bar (light green bars) in panels g-j, respectively.

| | | 532 nm | | 910 nm | |
|---|---|---|---|---|---|
| **Mass** | **Number, $c_n$** (Mm $\cdot$ m$^{-3}$) | $270 \pm 20$ | | $272 \pm 12$ | |
| | **Volume, $c_v$** ($10^{-12}Mm\cdot$ m$^{-3}\cdot$ m$^{-3}$) | $1.79 \pm 0.15$ | | $1.95 \pm 0.10$ | |
| **CCN-related** | | **c** (cm$^{-3}$) | **x** | **c** (cm$^{-3}$) | **x** |
| | CCN (0.13–35µm) | $0.01 \pm 10.68$ | $2.93 \pm 0.68$ | $0.45 \pm 3.33$ | $1.98 \pm 0.33$ |
| | GCCN (1–35µm) ($10^{-3}$) | $2.6 \pm 0.3$ | - | $2.7 \pm 0.3$ | - |
| | UGCCN (10–35µm) ($10^{-4}$) | $3.11 \pm 0.36$ | - | $2.57 \pm 0.14$ | - |

**Table 1.** Birch conversion parameters essential to convert the particle extinction coefficient of birch, $\alpha_{birch}$, at 532 nm and 910 nm into particle number and mass concentration. The mean values and standard error (SE) of the extinction-to-number, $c_n$, and extinction-to-volume, $c_v$ for birch pollen, are derived from in situ ICEMET observations considering the particle size range of 12–35 µm. The necessary conversion factors for the CCN-related estimations $c_{0.13-35\mu m,birch}$, $c_{1-35\mu m,birch}$, $c_{10-35\mu m,birch}$ and $x_{0.13-35\mu m,birch}$ (Eq. 5-7) are obtained according to Sect. 2.9.2 from NS+OPS+ICEMET, OPS+ICEMET and ICEMET observations, respectively.

|  |  | LR = 50 sr | | LR = 60 sr | | LR = 70 sr | | LR = 8 |
|---|---|---|---|---|---|---|---|---|
| **Mass** | **Number, $c_n$** $(Mm \cdot m^{-3})$ | $324 \pm 24$ | | $270 \pm 20$ | | $232 \pm 17$ | | $203 \pm$ |
|  | **Volume, $c_v$** $(10^{-12} Mm \cdot m^{-3} \cdot m^{-3})$ | $2.15 \pm 0.18$ | | $1.79 \pm 0.15$ | | $1.53 \pm 0.13$ | | $1.34 \pm$ |
| **CCN-related** |  | **c** $(cm^{-3})$ | **x** | **c** $(cm^{-3})$ | **x** | **c** $(cm^{-3})$ | **x** | **c** $(cm^{-3})$ |
|  | CCN (0.13–35μm) | 0.02±10.29 | 2.93±0.68 | 0.01±11.63 | 2.93±0.68 | 0.008±12.89 | 2.93±0.68 | 0.005±14.10 |
|  | GCCN (1–35μm) $(10^{-3})$ | 3.2±0.3 | - | 2.6±0.3 | - | 2.3±0.2 | - | 2.0±0.2 |
|  | UGCCN (10–35μm) $(10^{-4})$ | 3.73±0.43 | - | 3.11±0.36 | - | 2.26±0.31 | - | 2.23±0.39 |

**Table A1.** Effect of LR selection on the conversion factors at 532 nm.

| | | LR = 50 sr | | LR = 60 sr | | LR = 70 sr | | LR = 80 sr | |
|---|---|---|---|---|---|---|---|---|---|
| **Mass** | **Number, $\mathbf{c}_n$** (Mm $\cdot$ m$^{-3}$) | $326 \pm 15$ | | $272 \pm 12$ | | $233 \pm 10$ | | $204 \pm 9$ | |
| | **Volume, $\mathbf{c}_v$** ($10^{-12}Mm\cdot$ m$^{-3}\cdot$ m$^{-3}$) | $2.33 \pm 0.11$ | | $1.95 \pm 0.10$ | | $1.67 \pm 0.08$ | | $1.46 \pm 0.07$ | |
| **CCN-related** | | **c** (cm$^{-3}$) | **x** | **c** (cm$^{-3}$) | **x** | **c** (cm$^{-3}$) | **x** | **c** (cm$^{-3}$) | |
| | CCN (0.13–35μm) | 0.65±3.14 | 1.98±0.33 | 0.45±3.33 | 1.98±0.33 | 0.33±3.50 | 1.98±0.33 | 0.26±3.66 | 1.98± |
| | GCCN (1–35μm) ($10^{-3}$) | 3.4±0.3 | - | 2.7±0.3 | - | 2.3±0.2 | - | 2.0±0.2 | |
| | UGCCN (10–35μm) ($10^{-4}$) | 3.45±0.17 | - | 2.87±0.14 | - | 2.46±0.12 | - | 2.17±0.11 | |

**Table A2.** Effect of LR selection on the conversion factors at 910 nm.