# Peer review of "Lidar estimates of birch pollen number, mass and related CCN concentrations"

_EGUsphere, 2024_

## Author Comment (AC2)

*We would like to thank the reviewer for their valuable comments and suggestions. We have modified the manuscript to include the proposed changes along with step-by-step answers to their suggestions. We would also like to inform the reviewer of a major addition in the manuscript which is the inclusion of 532 nm wavelength utilizing the PollyXT observations.*

**Reviewer #1**

**This paper presents, for the first time, estimations of particle number, mass, CCN, giant CCN (GCCN), and ultra-giant CCN (UGCCN) concentrations derived from polarization lidar observations of birch pollen, going beyond the traditional distribution and classification of aerosol types in the atmosphere. Although there are still many aspects that need to be improved when compared to in-situ measurements at ground level, this study is deemed necessary from the perspective of extending lidar technology and making new attempts.**

**Therefore, it is judged appropriate for this paper to be published in the respective journal.**

**However, there is one important question. When discussing the particle size distribution and concentration, as shown in Figure 3, you compare the particle size of birch pollen using results obtained from the Burkard sampler and ICEMET. At this point, it is necessary to confirm whether the particle sizes reported by each instrument refer to aerodynamic particle size or geometric particle size. It seems that the particle size from the Burkard sampler is reported as the geometric particle size, but I am unsure about the particle size reported by ICEMET. If it is the aerodynamic particle size, it may require adjustments to compare the particle sizes derived from the two instruments.**

The reviewer raises a valid question. Both instruments report the geometric particle size. In the Burkard sampler, the pollen particles stuck on the tape and later are analyzed under the microscope. During this procedure, the tape is cut and mounted between two glasses submerged into a solution of gelvatol, ion exchanged water, glycerol and lactic acid to preserve the sample. In ICEMET, the particles are illuminated by a short laser light pulse and the resulting hologram is digitally sampled by a digital image sensor and the digital hologram is then numerically analyzed to calculate the size of the particles. The pulse length of the laser used is 50 ns, which is enough to freeze the moment of the objects to less than one pixel size in the hologram up to 30 m/s speed (Kaikkonen, Molkoselkä and Mäkynen, 2020). We have added the following sentence in Sect. 3.1.1 to support further the discussion: 'Also, both instruments sense the geometrical particle diameter, and therefore their diameters are directly comparable.'

Kaikkonen, V.A., Molkoselkä, E.O. & Mäkynen, A.J. A rotating holographic imager for stationary cloud droplet and ice crystal measurements. *Opt Rev* **27**, 205–216 (2020). https://doi.org/10.1007/s10043-020-00583-y

**Additionally, although it was mentioned that wind direction and wind speed were measured using a Doppler lidar, no results related to these measurements are presented in the paper. There is only a reference stating that it was used to identify the mixed layer and that data below 200 m were not used in the analysis. When comparing lidar measurements with in-situ measurements, as in Figure 7, it seems necessary to check whether meteorological conditions, especially wind speed or diffusion coefficients at different altitudes, had any effect. In this context, Doppler lidar data could be utilized.**

The MLH was selected as an indicator of similar aerosol conditions between the ground and the 200m level. The MLH was estimated using a threshold in the Turbulent Kinetic Energy (TKE) dissipation rate profile which is informative of the intensity of turbulence for a given flow. Having said that, whether the aerosol size distribution is similar between the surface and the 200m height level is multi-dimensional and it depends on the particle size and wind conditions. An indicator of whether pollen is distributed equally in the vertical direction would be the shape of the particle depolarization ratio (PDR). Unfortunately, this information is missing close to the surface and conclusions relying on the shape of the PDR profile can be made for heights above 200 m. The graph below shows similar information to figure 5 for the number concentration and birch extinction coefficient but without filtering the MLH nor the dust/bc cases. We can see that the MLH presents an efficient way to facilitate comparisons between the two height levels (panels a) and b)). Furthermore, data points following linear relationship present, on average, smaller wind speed difference between the surface and the elevated layer compared to those with similar wind speed conditions between the two height levels (panels c) and d)). Note that we have added PollyXT observations, therefore, the same information is additionally presented at 532 nm where the observations are available above 400 m due to the overlap limitation of that instrument. The highest birch concentration on site, is represented with the topmost point in all four panels. It was observed by Burkard instrument during the 12[th] of May 2021 at 8 UTC (7-9 UTC). We see that in 532~nm this point deviates from the linearity, and it can be due to the transitioning of the boundary layer which resulted to averaging heterogeneous aerosol layers together with the wavelength sensitivity to the aerosol particle size population. To this direction, we have added to Fig.7 the wind speed-direction at three levels (surface, 200m and 400m), the times that the mixing layer height was above the 200m (400m) height level as light green bar (dark green bars) as indicated by the HALO Doppler lidar and enhanced the discussion by adding the following sentences. 'In fact, the smaller the wind speed difference between the surface and the elevated layer is the better the agreement between these two height levels. In turn this implies that, during unstable atmospheric conditions, higher discrepancies between the lidar- and in situ-estimated quantities are anticipated, due to the long temporal averaging of non-uniform aerosol layers together with the sensitivity of the specific wavelength to the aerosol particle size distribution.' The updated figure 7 can be also found below.

[Figure]

**Other revisions or questions are as follows:**

**1. The CL 61 instrument is said to have a full overlap at 300 m, but in the study, data measured at 200–250 m are analyzed. Please provide an explanation for this.**

The 300 m is the height where the laser beam is mirrored in the field of view of the instrument. For quantities that are determined from signal ratios such as the depolarization ratio, the height of complete overlap is not as essential as for the separate detection channels, for example the attenuated backscatter coefficient. At 200 m height about 89-90% of the beam is mirrored (see figure below) and it is a compromise for having as close to the ground observations for the calibration of the lidar and assure quality assured signals. The overlap functions presented here are from 2023 and 2024 observations as the overlap function was not stored in the files in 2021 data. To this direction, the graph below shows five 1h temporally averaged profiles of the attenuated backscatter (att. bsc) and volume depolarization ratio (VDR) up to 0.6 km in height. The 200 m level is marked with a horizontal line. The 5 profiles are during clear skies with varying aerosol load and as minimal aerosol structure in the vertical direction as possible. A nested zoomed view is presented for the first 3 cases for the abovementioned quantities. We can see that there are structural similarities in the shape of the signal regardless of the aerosol load below 200 m. There are two local maxima present at about 50 m and 140 m. The VDR is less affected by these and information below 200 m can be possibly used for either optical property on a case-by-case scenario (e.g. for the high aerosol load case (yellow)). Below the selected height, information should be used with cautious and additional corrections may needed to be applied to derive useful information. Such corrections are outside of the scope of this paper.

[Figure]

**2. The lowest observation altitude for the Doppler lidar is indicated, but the highest observation altitude is not. Please also indicate the highest observation altitude.**

Thank you for your comment. We have added the following text to the manuscript: 'The minimum usable range of the instrument is 90m, as the lower range gates are affected by the outgoing pulse, and the maximum range is 9.6km above ground level'

---

## Author Comment (AC3)

*We would like to thank the reviewer for their valuable comments and suggestions which have improved this manuscript. We have modified the manuscript to include the proposed changes along with step-by-step answers to their suggestions. We would also like to inform the reviewer of a major addition in the manuscript which is the inclusion of 532 nm wavelength utilizing the PollyXT observations.*

**Reviewer #3**

**The manuscript presents a multi-disciplinary study that combines in-situ and remote sensing data to provide newly derived extinction to concentration conversion factors for Birch Polen particles at 905 nm. Such factors are of high importance in the lidar and ceilometer communities as they can be used for the monitoring of the vertical distribution of pollen concentration that is in-turn important for public-health and for studying aerosol-cloud interactions. The manuscript is well structured and clear. The techniques are described sufficiently. I recommend the publication of the manuscript after some minor revisions according to the comments below.**

**Lines 98-100: Is this uncertainty related only to the calibration? is the uncertainty due to lidar ratio biases taken into account too? If not please add the anticipated uncertainty if the lidar ratio is not 60 sr.**

Thank you for your comment. This uncertainty refers to the particle backscatter coefficient. The lidar ratio selection introduces negligible uncertainty in the boundary layer for the forward inversion and thus omitted. This is visualized in the example below, where a forward Klett inversion has been performed for a variety of lidar ratios (LRs). In the backward Klett inversion the LR induces greater variability as shown in Shang et al., (2021). Note that all other parameters are kept constant except the LR.

Shang, X., Mielonen, T., Lipponen, A., Giannakaki, E., Leskinen, A., Buchard, V., Darmenov, A. S., Kukkurainen, A., Arola, A., O'Connor, E., Hirsikko, A., and Komppula, M.: Mass concentration estimates of long-range-transported Canadian biomass burning aerosols from a multi-wavelength Raman polarization lidar and a ceilometer in Finland, Atmos. Meas. Tech., 14, 6159–6179, https://doi.org/10.5194/amt-14-6159-2021, 2021.

[Figure]

**Lines 101-102: Is an overlap correction being applied down to 300 m (or 200 m). If yes, which is the real distance of full overlap where the correction starts? Please specify.**

Thank you for your comment. Yes, an overlap correction is automatically included/performed in the attenuated backscatter coefficient provided by the manufacturer/instrument. The correction is applied from the lowest height bin up to the height of full overlap. Below are two overlap functions from 2023 and 2024 observations as the overlap function was not stored in CL61 files in 2021. An example of five 1h temporally averaged profiles of the attenuated backscatter (att. bsc) and volume depolarization ratio (VDR) up to 0.6 km in height is also shown. The 200 m level is marked with horizontal line.  Regarding the overlap function, at 200 m about 89-90% of the beam is mirrored by the telescope. Regarding the 5 profiles, these are taken during clear skies with varying aerosol load and as minimal aerosol structure in the vertical direction as possible. A nested zoomed view is presented for the first 3 cases for the abovementioned quantities.  We can see that there are structural similarities in the shape of the signal regardless of the aerosol load below 200 m. There are two local maxima present at about 50 m and 140 m. The VDR is less affected by these and information below 200 m can be possibly used for either optical property on a case-by-case scenario (e.g. for the high aerosol load case (yellow)). Below the selected height, information should be used with cautious and additional corrections may needed to be applied to derive useful information. Such corrections are outside of the scope of this paper.

[Figure]

**Lines 179-181: In section 2.5.1 a size limit of 5.3 μm is reported for ICEMET. This seems to be in conflict with the 12-35 μm range mentioned here. The different values probably refer to different types of size distribution (number concentration and volume concentration) but this is not clearly mentioned. Please add a brief explanation.**

Thank you for your comment. Although ICEMET provides the aerosol size distribution between 5.3 and 200 um and adding the NS/OPS aerosol size distributions we have the aerosol distributions for even smaller particle sizes, for the number and mass concentrations we have utilized the size range between 12 and 35 μm. The selection of 12-35 μm was chosen as representative of birch pollen particles, thus, the relationship between extinction and number/mass is more straightforward for this aerosol type. Nevertheless, in Sect. 4 (Lines 367-374) we discuss the selection of the lower size particle limit and its effect in the estimated conversion factors. More specifically, we mention that:

'For estimating the mass concentration, even if smaller coarse birch pollen particles or other biological material are present in the atmosphere (> 2.5 μm diameter), the uncertainty in the $c_v$ factor is in the order of 5 % with a re–estimated $c_v$ of 1.90 ± 0.10, which is within the uncertainty range of the 12–35 μm size range. Assuming an AERONET equivalent particle size range (1.2–30 μm diameter), $c_v$ of 1.84 ± 0.08 is obtained which presents a 6 % discrepancy from the 12–35 μm size range. In comparison, when using the AERONET method (level 1.5), a $c_v$ of 1.24 ± 0.06 is estimated. This is not the case for the number concentration estimation in which the inclusion of aerosol particles above 2.5 μm leads to an order of magnitude higher $c_n$ factor than the 12–35 μm size range, with the relationship between volume concentration and extinction not to be linear anymore.

**Lines 207-208: What are the two components? Is it pollen and water solubles (sulfate, nitrate organics)?**

We assume a mixture of birch pollen (non-spherical component) and background aerosols which present a very low particle depolarization ratio (spherical component) (Shang et al., 2020; Bohlmann et al., 2019) for the decomposition method. Having said that, there are studies that have explored the chemical composition of the aerosol population in the area. For example, in Portin et al., (2014) sulfate, nitrate, ammonium and organics were all present at Kuopio (20 km from Vehmäsmäki station) where the inorganic to total ratio was about 42 (%). The following text has been added to the manuscript: 'Portin et al. (2014) have explored the chemical composition of the aerosol population in the area and found that sulfate, nitrate, ammonium and organics are present at Kuopio, about 20~km from Vehmäsmäki station, where the inorganic to total ratio was about 42%.'

Portin, H., Leskinen, A., Hao, L., Kortelainen, A., Miettinen, P., Jaatinen, A., Laaksonen, A., Lehtinen, K. E. J., Romakkaniemi, S., and Komppula, M.: The effect of local sources on particle size and chemical composition and their role in aerosol–cloud interactions at Puijo measurement station, Atmos. Chem. Phys., 14, 6021–6034, https://doi.org/10.5194/acp-14-6021-2014, 2014.

**Lines 205-207: From which instrument/wavelength? Is this still the ceilometer?**

The sentence relates to the methodology of retrieving the extinction-to-number (or mass) conversion factor and refers to any instrument/wavelength capable of providing the particle extinction coefficient. Since we have now added PollyXT lidar measurements and therefore, 532 nm wavelength was added, we have modified the sentence as follows. 'The second required parameter for the $c_v$ ($c_n$) calculation is the α for the specific aerosol type. The birch extinction coefficient, $α_{birch}$, was derived by polarization

lidar observations based on the backward (forward) Klett–Fernald inversion method for PollyXT (CL61) observations, respectively, and the birch component separation method from Tesche et al. (2009)'

**Lines 211-212: Here it is implied, that a MLH ensures well mixed conditions up to 200 m so that the in-situ and the remote sensing retrievals can be combined. Is this the case? Please elaborate more as this is a key part of this study.**

The MLH was selected as an indicator of similar aerosol conditions between the ground and the 200m level. The MLH was estimated using a threshold in the Turbulent Kinetic Energy (TKE) dissipation rate profile which is informative of the intensity of turbulence for a given flow.  Having said that, whether the aerosol size distribution is similar between the surface and the 200m height level is multi-dimensioned and it depends on the particle size and wind conditions. One more factor to be considered is the 2h temporal averaging that can affect the aerosol distribution, especially during the transition times of the boundary layer and can introduce dissimilarities in the concentrations between the lidars and the in-situ instruments. An indicator of whether pollen is distributed equally in the vertical direction would be the shape of the particle depolarization ratio (PDR). Unfortunately, this information is missing close to the surface and conclusions relying on the shape of the PDR profile can be made for heights above 200 m or higher, depending on the lidar. The graph below shows similar information to figure 5 for the number concentration and birch extinction coefficient but without filtering the MLH nor the dust/bc cases. We can see that the MLH presents an efficient way to facilitate comparisons between the two height levels (panels a) and b)). Furthermore, data points following linear relationship present smaller wind speed difference between the surface and the elevated layer compared to those with similar wind speed conditions between the two height levels (panels c) and d)). Note that we have added PollyXT observations, therefore, the same information is additionally presented at 532 nm where the observations are available above 400 m due to the overlap limitation. The highest birch concentration on site, is represented with the topmost point in all four panels. It was observed by Burkard instrument during the 12[th] of May 2021 at 8 UTC (7-9 UTC). We see that in 532~nm this point deviates from the linearity, and it can be due to the transitioning of the boundary layer which resulted to averaging heterogeneous aerosol layers together with the wavelength sensitivity to the aerosol particle size population. To this direction, we have added to Fig.7 the wind speed-direction at three levels (surface, 200m and 400m), the times that the mixing layer height was above the 200m (400m) height level as light green bar (dark green bars) as indicated by the HALO Doppler lidar and enhanced the discussion by adding the following sentences. 'In fact, the smaller the wind speed difference between the surface and the elevated layer is the better the agreement between these two height levels. In turn this implies that, during unstable atmospheric conditions, higher discrepancies between the lidar- and in situ-estimated quantities are anticipated, due to the long temporal averaging of non-uniform aerosol layers together with the sensitivity of the specific wavelength to the aerosol particle size distribution.' The updated figure 7 can be also found below.

[Figure]

**Lines 211-212: Does the birch share correspond to volume or number concentration fraction? Please specify. Recommendation: It would be interesting to know how does 90% contribution to the volume (or number) concentration translate to extinction (or backscatter) contribution? This can be estimated with the same methodology described here.**

Thank you for your comment. The birch share in Lines 211-212 corresponds to the number concentration, and it refers to the share of birch pollen compared to the rest pollen types without accounting the contribution of other aerosols in the mixture. In this way, we know that in the considered cases, the effect of birch pollen is studied. Then, during the decomposition lidar method,

the share of birch pollen to the total particle backscatter coefficient (birch share from lidar) can be determined using the PDR value of birch which for the 910nm is 0.23 (Filioglou et al., 2023). To this direction, the birch particle backscatter coefficient to the birch share from the lidar can be also estimated using the decomposition method. The color indicates the share of PM10 to the total mass using the method described in Filioglou et al., (2023). Specifically, the share of $PM_{10}$ in the aerosol mixture was calculated as follows: $100 \cdot PM_{10}$ $(PM_{10} + PM_{pollen})$. We have used a birch pollen diameter of 25~μm for the conversion of number pollen concentration to mass. The size indicates the number concentration of birch pollen from Burkard estimations. Note that all cases are included here without filtering the data with the MLH or for bc/dust contributions. We can see that the estimated birch share from the lidar and the share of PM10 are anticorrelated where the higher the contribution of birch pollen is in the lidar observations (implying higher PDR), the less the contribution of PM10 is in the aerosol mixture. This is the behaviour we expect. Then, mixtures of birch with other unspherical aerosol particles, can deviate from this trend. For example, in such cases SPPs of birch pollen or/and other biological material (e.g., spores, fungi, algae etc) having different PDR than birch pollen with not know concentration can be present.

[Figure]

**Lines 215-218: Is the LR of birch particles 60 sr or is this a general climatological value? Please specify and add a reference here.**

Thank you for your question. The LR is a climatological value as reported in Bohlmann et al. (2019) and Shang et al. (2020, 2022) studies where birch was present. The following references have been added. Also, we have added two tables, one for each wavelength, presenting a sensitivity study to the selection of the LRs to the estimated conversion factors.

Bohlmann, S., Shang, X., Giannakaki, E., Filioglou, M., Saarto, A., Romakkaniemi, S., and Komppula, M.: Detection and characterization of birch pollen in the atmosphere using a multiwavelength Raman polarization lidar and Hirst-type pollen sampler in Finland, Atmos. Chem. Phys., 19, 14559–14569, https://doi.org/10.5194/acp-19-14559-2019, 2019.

Shang, X., Giannakaki, E., Bohlmann, S., Filioglou, M., Saarto, A., Ruuskanen, A., Leskinen, A., Romakkaniemi, S., and Komppula, M.: Optical characterization of pure pollen types using a multi-wavelength Raman polarization lidar, Atmos. Chem. Phys., 20, 15323–15339, https://doi.org/10.5194/acp-20-15323-2020, 2020.

Shang, X., Baars, H., Stachlewska, I. S., Mattis, I., and Komppula, M.: Pollen observations at four EARLINET stations during the ACTRIS-COVID-19 campaign, Atmos. Chem. Phys., 22, 3931–3944, https://doi.org/10.5194/acp-22-3931-2022, 2022.

**Lines 215-218: According to section 2.2 the ceilometer full overlap is 300 m. How much is the systematic uncertainty due to the incomplete overlap at 200 m? Is an overlap correction being applied between 200 and 300 m. Please specify**

We would like to refer the reviewer to the second comment as we have combined the answers from this question there.

**Lines 221-222: Please don't forget to mention which concentration is being used each time, number or volume?**

Thank you for your suggestion. We have added the missing information in places where it was missing.

**Lines 236-238: Suggestion: Move this sentence higher up in this section so that the readers can follow more easily the discussion.**

Moved according to reviewer's suggestion.

**Lines 225-226: As this is a multi-disciplinary study, it would be beneficial to provide a brief explanation of what kappa-value is for non CCN experts.**

Thank you. We have added the following text: 'At this supersaturation, Mikhailov et al. (2021) found the hygroscopicity of birch pollen particles, kappa–value, to be k = 0.13 ± 0.02 and an estimation of the activated particles can be made according to…..'

**Lines 243-250: The factor f_ss, birch is never introduced. Please add a description.**

Added according to reviewer's suggestion.

**Lines 240-261: Most of the factors/variables here were never properly introduced. Please add a discription of what each factor corresponds too. The subscripts are not sufficient to deduce the variable's role.**

Added according to reviewer's suggestion.

**Lines 284-285: Are sea salt particles expected at the site?**

According to Portin et al., (2009) and Leskinen et al., (2012), the site is a receptor of air masses originating both from continental and marine environment.

Portin, H. J., Komppula, M., Leskinen, A. P., Romakkaniemi, S., Laaksonen, A., & Lehtinen, K. E. J. (2009). Observations of aerosol–cloud interactions at the Puijo semi-urban measurement station. Boreal Environment Research Publishing Board

Leskinen, A., Arola, A., Komppula, M., Portin, H., Tiitta, P., Miettinen, P., Romakkaniemi, S., Laaksonen, A., and Lehtinen, K. E. J.: Seasonal cycle and source analyses of aerosol optical properties in a semi-urban environment at Puijo station in Eastern Finland, Atmos. Chem. Phys., 12, 5647–5659, https://doi.org/10.5194/acp-12-5647-2012, 2012.

---

## Author Comment (AC4)

*We would like to thank the reviewer for their valuable comments and suggestions which have improved this manuscript. We have modified the manuscript to include the proposed changes along with step-by-step answers to their suggestions.*

**Reviewer #2**

**The authors present a study to retrieve extinction-to-microphysics conversion parameters of pollen aerosol. These conversion parameters can be applied to profiles of pollen aerosol-optical properties derived by polarization lidar observations in order to estimate profiles of cloud-relevant properties, in this case pollen cloud condensation nuclei concentrations. The presented method is analogous to the well-established POLIPHON (Polarization Lidar Photometer Networking) method, however it is using ceilometer with polarization capability and ground-based in situ observations instead of long-term AERONET sun photometer observations to link particle extinction with particle size distributions. This link is then exploited to retrieve abovementioned conversion parameters.**

**Such novel (in terms of aerosol type) conversion parameters are always sought-after for the application to lidar profiles all over the world, either from ground-based long-term observations or networks, or from space-borne lidars, which allows to retrieve climatologies of microphysical and cloud-relevant properties. Furthermore, very special observations periods at specific sites equipped with appropriate instrumentation are needed to perform such correlation studies.**

**Therefore, the manuscript is suited for publication in Atmospheric Chemistry and Physics and can be published almost as is after addressing rather minor comments/questions listed below.**

**Major comment:**

**My main question following hereafter is with regard to wavelengths and wavelength conversion. 355 and 532 nm are much more common lidar wavelengths (of ground-based but also space-borne systems, which you mention in your conclusion), especially concerning polarization capability. I believe it would be worth adding some statements to the manuscript regarding this topic.**

Thank you for your suggestion. In addition to the 910 nm wavelength from CL61 ceilometer, we have added PollyXT observations and provided the number, mass and CCN-related conversion factors at 532 nm, as well. Similar to CL61 ceilometer observations, we have performed the Klett inversion (backward) to PollyXT 532 nm observations since the combination of background light, overlap height and boundary layer height impairs the Raman retrieval and limits the availability and diversity of cases in terms of pollen loads. Note that observations at 355 nm are omitted due to the high overlap which is at 800 m which together with the birch PDR which is close to the background PDR in this wavelength, introduces high uncertainty in the retrievals and limits the availability of cases. As an outcome, a new section has been added to the manuscript and Figs. 5, 6 and 7 have been updated.

**There is a PollyXT is at this site (Line 75)? Does it have near-range Raman capability? Its data are obviously not used in this study. Is there a specific reason for that? I see that there would be potentially two problems: first, the pollen being present at very low ranges below the overlap region of that larger lidar; and second, potentially the polar day at these high latitudes in summer (already in May?) impairs Raman retrieval due to background light.**

**At least in e.g., Bohlmann et al. (2019, 2021), this seemed to be not a significant problem, and at least backscatter and depolarization profiles down to low ranges at low vertical averaging would be retrievable by Raman method (at nighttime), even lower using near-range capability, including extinction, however missing depolarization, if I understand the (obviously various) PollyXT setups correctly.**

**Could/did you compare nighttime Raman extinctions (at 355 and 532 nm) to the Klett-retrieved extinction (from backscatter, at 910 nm)? What about somehow verify the choice of the lidar ratio (of 60 sr at 910 nm) or discuss the sensitivity of the retrieved conversion parameters on that choice?**

**As Bohlmann et al. (2019, 2021), Shang et al. (2020, 2022) and Filioglou et al. (2023) nicely discussed pollen lidar ratios, depolarization ratios, however, for 355 and 532 nm wavelength, and maybe most importantly in this context here, also some Ångström exponents (also backscatter-related 532/1064 and 532/910, respectively), it would be very beneficial at least to discuss/provide some more literature values or methods/ideas/suggestions helping in the wavelength conversion of your results.**

The PollyXT version installed in Vehmasmäki does include a near-field Raman capability, but polarization measurements are available only in the far-field view. Related to the Raman inversion, pollen activity usually peaks during daytime which prohibits the use of Raman channels while during nighttime, the boundary layer height is low, limiting the availability and diversity of cases in terms of pollen loads. An added difficulty is the geographical location of the station which from May and on there is no night with the sun staying between 6 and 12 degrees below the horizon at the darkest hours. In fact, the PollyXT cases added in the manuscript are during daytime (between 06 and 16 UTC) with only 2 cases at 18 UTC (21 local time).

Having said that, in the link below you can find Raman retrievals for the 12[th] of May 2021 for PollyXT instrument available online at PollyNET. Targeting the darkest hour at the station, which is around 21-23 UTC (the last two rows of graphs in the provided link), we can see lidar ratios fluctuating between 50 and 80 sr in the lowest 1 km. It is evident that the Raman-shifted wavelength at 607 nm is noisier compared to the 387 nm. A similar solution but applying a vertical smoothing window of 21 bins which translates to 157 m and averaging between 21 and 23 UTC, can be found below. In addition to the PollyXT optical products, the CL61 ceilometer particle backscatter coefficient and particle depolarization ratio have been added. We have also used the decomposition method to derive the ratio of birch pollen in the mixture. Accounting for the overlap and vertical smoothing, profiles above 400-500 m can be considered. At 500 m, the particle depolarization ratio value is 0.20 at 532 nm (0.13 at 910nm) and Angstrom exponents are in the range of 0.45-0.86 pointing to mixtures of pollen and background aerosols. This can also be seen in the birch share profiles. At the moment this is the best guess of the lidar ratio as already mentioned in the studies above but surely more efforts should be put for deriving the lidar ratio of birch pollen.

The lidar ratio selection introduces negligible uncertainty in the boundary layer for the forward inversion. This is visualized in the example below, where a forward Klett inversion has been performed for a variety of lidar ratios (LRs). In the backward Klett inversion the LR induces greater variability as shown in Shang et al., (2021). Note that all other parameters are kept constant except the LR. Since the lidar ratio is decisive for converting the particle backscatter to extinction coefficient regardless of the inversion method, we have included a sensitivity study summarized in the tables below which show how the conversion factors change depending on the selection of the lidar ratio. The tables have been included in the manuscript as well.

[Figure]

Shang, X., Mielonen, T., Lipponen, A., Giannakaki, E., Leskinen, A., Buchard, V., Darmenov, A. S., Kukkurainen, A., Arola, A., O'Connor, E., Hirsikko, A., and Komppula, M.: Mass concentration estimates of long-range-transported Canadian biomass burning aerosols from a multi-wavelength Raman polarization lidar and a ceilometer in Finland, Atmos. Meas. Tech., 14, 6159–6179, https://doi.org/10.5194/amt-14-6159-2021, 2021.

[Figure]

Figure: a) Raman-estimated particle backscatter coefficients at 355, 532 and 1064nm from PollyXT lidar. For 355 and 532 nm, the near-field (NF) signals have been used. Particle backscatter coefficient at 910 nm from CL61 ceilometer is shown in solid orange line. b) Particle extinction coefficient at 355 and 532 nm. c) Lidar ratio profiles at 355 and 532 nm. d) Backscatter-related Angstrom exponents. e) Particle depolarization ratio at 532 and 910 nm. f) Estimated birch share in the total backscatter from 532 nm and 910 nm using the decomposition method.

| | | LR = 50 sr | | LR = 60 sr | | LR = 70 sr | | LR = 80 sr | |
|---|---|---|---|---|---|---|---|---|---|
| **Mass** | **Number, $c_n$** $(Mm \cdot m^{-3})$ | 324 ± 24 | | 270 ± 20 | | 232 ± 17 | | 203 ± 15 | |
| | **Volume, $c_v$** $(10^{-12} Mm \cdot m^{-3} \cdot m^{-3})$ | 2.15 ± 0.18 | | 1.79 ± 0.15 | | 1.53 ± 0.13 | | 1.34 ± 0.11 | |
| | | **c** $(cm^{-3})$ | **x** | **c** $(cm^{-3})$ | **x** | **c** $(cm^{-3})$ | **x** | **c** $(cm^{-3})$ | **x** |
| **CCN-related** | **CCN (0.13-35 um)** | 0.02 ± 10.29 | 2.93 ± 0.68 | 0.01 ± 11.63 | 2.93 ± 0.68 | 0.008 ± 12.89 | 2.93 ± 0.68 | 0.005 ± 14.10 | 2.93 ± 0.68 |
| | **GCCN (1-35 um) $(10^{-3})$** | 3.2 ± 0.3 | - | 2.6 ± 0.3 | - | 2.3 ± 0.2 | - | 2.0 ± 0.2 | - |
| | **UGCCN (10-35 um) $(10^{-4})$** | 3.73 ± 0.43 | - | 3.11 ± 0.36 | - | 2.26 ± 0.31 | - | 2.23 ± 0.39 | - |

Table 1. Birch conversion parameters essential to convert the particle extinction coefficient of birch, $\alpha_{birch}$, at 532 nm into microphysical properties for different lidar ratios (LR).

| | | LR = 50 sr | | LR = 60 sr | | LR = 70 sr | | LR = 80 sr | |
|---|---|---|---|---|---|---|---|---|---|
| **Mass** | **Number, $c_n$** $(Mm \cdot m^{-3})$ | 326 ± 15 | | 272 ± 12 | | 233 ± 10 | | 204 ± 9 | |
| | **Volume, $c_v$** $(10^{-12} Mm \cdot m^{-3} \cdot m^{-3})$ | 2.33 ± 0.11 | | 1.95 ± 0.10 | | 1.67 ± 0.08 | | 1.46 ± 0.07 | |
| | | **c** $(cm^{-3})$ | **x** | **c** $(cm^{-3})$ | **x** | **c** $(cm^{-3})$ | **x** | **c** $(cm^{-3})$ | **x** |
| **CCN-related** | **CCN (0.13-35 um)** | 0.65 ± 3.14 | 1.98 ± 0.33 | 0.45 ± 3.33 | 1.98 ± 0.33 | 0.33 ± 3.50 | 1.98 ± 0.33 | 0.26 ± 3.66 | 1.98 ± 0.33 |
| | **GCCN (1-35 um) $(10^{-3})$** | 3.4 ± 0.3 | - | 2.7 ± 0.3 | - | 2.3 ± 0.2 | - | 2.0 ± 0.2 | - |
| | **UGCCN (10-35um) $(10^{-4})$** | 3.45 ± 0.17 | - | 2.87 ± 0.14 | - | 2.46 ± 0.12 | - | 2.17 ± 0.11 | - |

Table 2. Birch conversion parameters essential to convert the particle extinction coefficient of birch, $\alpha_{birch}$, at 910 nm into microphysical properties for different lidar ratios (LR).

Automatically Raman-retrieved profiles of backscatter and extinction coefficients and lidar ratios from both near- and far-field can be seen here:

https://polly-tmp.tropos.de/datavis/location/3/25/2/?dates=[2021-05-12T00:00:00,2021-05-13T00:00:00]

**Minor comments:**

**Line 58: I urgingly suggest to add e.g., in front of He et al. (2023), or add other studies. It has been done many more times and before than only He et al. (2023).**

Thank you for your suggestion. We have added the following studies:

Shinozuka, Y., Clarke, A. D., Nenes, A., Jefferson, A., Wood, R., McNaughton, C. S., Ström, J., Tunved, P., Redemann, J., Thornhill, K. L., Moore, R. H., Lathem, T. L., Lin, J. J., and Yoon, Y. J.: The relationship between cloud condensation nuclei (CCN) concentration and light extinction of dried particles: indications of underlying aerosol processes and implications for satellite-based CCN estimates, Atmos. Chem. Phys., 15, 7585–7604, https://doi.org/10.5194/acp-15-7585-2015, 2015.

Mamouri, R. E. and Ansmann, A.: Estimated desert-dust ice nuclei profiles from polarization lidar: methodology and case studies, Atmos. Chem. Phys., 15, 3463–3477, https://doi.org/10.5194/acp-15-3463-2015, 2015.

Mamouri, R.-E. and Ansmann, A.: Potential of polarization lidar to provide profiles of CCN- and INP-relevant aerosol parameters, Atmos. Chem. Phys., 16, 5905–5931, https://doi.org/10.5194/acp-16-5905-2016, 2016.

Ansmann, A., Mamouri, R.-E., Hofer, J., Baars, H., Althausen, D., and Abdullaev, S. F.: Dust mass, cloud condensation nuclei, and ice-nucleating particle profiling with polarization lidar: updated POLIPHON conversion factors from global AERONET analysis, Atmos. Meas. Tech., 12, 4849–4865, https://doi.org/10.5194/amt-12-4849-2019, 2019.

**Line 133/Fig. 3: Is this software what is used to produce these blue lines in Fig. 3? It is not clear from the caption what these lines are (indicated length and text are written in a much too small font). Please, include it.**

That is correct. The lines indicate the geometrical diameter of the birch pollen grain where up to 40 diameters per sample were measured in order to extract the statistical population of the birch pollen. The following information has been added to the caption: 'The Olympus cellSens Entry imaging software was used to mechanically measure the diameters of the particles (blue lines) identified as birch pollen grains for up to 40 individual grains.'

**Line 360: Concerning the log-log regression in the relationship between number concentration and extinction, usually Shinozuka et al. (2015, https://doi.org/10.5194/acp-15-7585-2015) is given as a reference. At least log-log regression is nowadays commonly done when using AERONET to retrieve conversion parameters (for number concentrations), however, usually rather for the smaller particles (r>50 nm and 100 nm, i.e., marine, continental/pollution) than for the larger particles (r>250 nm, i.e., dust) (e.g., Mamouri and Ansmann, 2016).**

Indeed! We have revised Section 2.8.2 and replot Figures 6b-c and 7 regarding the GCCN and UGCCN. We have kept the log-log relationship for the CCN retrievals only.

**It is correctly stated in the abstract and in Sect. 3.2.1. that the conversion parameters are only retrieved from 2021 measurements. This becomes clear as ICEMET data were only available in**

**2021 and 2023 (Line 160), but missing in 2022, and on the other hand, ceilometer data were not available in 2023 (Line 104). It might be helpful to make this fact a bit clearer by explicitly stating that, despite the fact that impressive multiyear pollen observations are available, the concurrent measurements are only available for 2021.**

We have added the following statement: 'Despite the multiyear pollen observation availability on site, the conversion factors for the number/mass concentrations from PollyXT are extracted from 2-year observations during 2021 and 2023 while equivalent factors from CL61 ceilometer consider 2021 observations due to the instrument availability of the sensors involved. For the CCN–related conversion factors, the datasets used for CCN and GCCN are during 2021 for both lidars while UGCCN dataset for PollyXT includes both 2021 and 2023. For CL61, UGCCN conversion factor is extracted from the 2021 dataset'

**Line 416: Here, I would recommend to repeat that statement from the abstract regarding ceilometer-networks. Of course, they are meant inclusively in lidar-networks, but one could highlight the advantage of the dense coverage of such systems/networks, even if they are not yet throughout equipped with polarization capability.**

Thank you for your suggestion. We have added the suggested statement.

**Also Line 416: There is no reason to restrict it to space-borne backscatter lidars. I see that this could be meant as the minimum requirement, but I suggest to just state "space-borne lidars".**

Corrected according to reviewer's suggestion.

**Minor comments on style and spelling:**

**There are many cases of missing dots after Fig. etc., not abbreviated Section or Figure(s), missing s for plural Equations/Figrues, one missing ~ or \, after Sect. (Line 100-101), comma after "e.g.". Generally, it would also be beneficial to make figure and section references clickable by using \ref{label}.**

Thank you for your suggestions, we have now corrected the manuscript accordingly.

**There are many hyphens mixed up. Use "--" for intervals, maybe for things like aerosol--cloud interaction, but not for things like aerosol-type-dependent.**

Thank you for your comment. Corrected accordingly.

**In equations/formulas, generally, do not use italics for indices, use for example $_{\mathrm{index}}$. I suggest to do the same as well for units, only put the exponents in math mode cm$^{-3}$ (consistently), and use \textmu for micrometer.**

Thank you for your suggestions, the equations and formulas have been corrected accordingly.

**Line 103: I suggest to add "those": "therefore those data were omitted". The sentence before is also quite incomprehensible. I suggest to change it like that by adding two commas: "We only considered observations during 2021 and 2022, since during the birch pollen period in May 2023, the instrument experienced…".**

Thank you for your suggestion. The text was corrected accordingly. Also, the previous sentence has been reformed for clarity.

**Line 114: use \citep[e.g.,][]{Vakkari_2015} to avoid double )).**

Thank you, corrected as suggested.

**Line 139: Kaikkonen et al. (2020) refers to ICEMET only? Then, also add it like this: \citep[hereafter ICEMENT; University of Oulu, Finland][]{Kaikkonen_2020}.**

Thank you for your suggestion. Yes, it referes to ICEMET only. Corrected accordingly.

**Line 207: Tesche et al. (2009) with \citet{} instead of \citep{}.**

Corrected as suggested.

**Line 374: "CCN concentrations […] are caused" instead of "is caused".**

Corrected as suggested.

**Line 378: "model-based" instead of "modeled--based".**

Corrected as suggested.

**Line 379: Wozniak et al. (2018) with \citet{} instead of \citep{}.**

Corrected as suggested.

**Line 388: "coastal" instead of "Coastal".**

Noted and corrected.

**Fig. 1: Last access date of that web page has to be stated. And a comma added: "In close proximity,".**

Last access information added.

**Figs. 2 and 7: I would state "number concentration" and "Mass concentration", respectively, at least in the captions.**

Captions are corrected as suggested.

**Fig. 4: Add spaces or hyphens between 2 and h, "2-h" or "2 h" (actually, in the whole manuscript). Furthermore, I would not write "perfect fit" but "1:1 reference line", "identity line" or "line of equality".**

Noted and correct in Fig. 4 and throughout the manuscript.

**References**:

***Generally, it would be useful to have the references clickable in the text (copernicus class should do this automatically, though).***

***Generally, consistently use either full journal name or abbreviation (e.g., Lines 496, 520, 546, 556, 654).***

***Generally, remove all double https://doi.org/ in https://doi.org/https://doi.org/.***

Thank you for the thorough review. We have removed the repetition of  https://doi.org/ and opted for the full journal names.

**Lines 454 and 455: some strange line breaks (in Baars et al., 2016).**

Noted and corrected.

**Lines 466, 507, 587, 590, 593:  no space between pages and hyphen.**

Noted and corrected.

**Line 479: In Buters et al. (2018), Galán is not correctly parsed.**

Corrected as suggested.

**Line 486: The degree sign in the title is missing (Cholleton et al., 2022).**

Added as suggested.

**Lines 521, 522: Why are two urls given? And last access date of both should be given.**

Noted and last access date added.

**Lines 553, 554 and 579, respectively: Last access dates have to be stated in the same format.**

Noted and corrected.

**Line 561: In Lewis and Schwartz (2004), the doi seems to be not working (besides the issue with double https://doi.org/), even though it is stated as well on this url: https://agupubs.onlinelibrary.wiley.com/doi/abs/10.1002/9781118666050.ch5. I do not know how to deal with this issue. I leave it to the copy editors.**

Noted and corrected.

**Line 627: Journal name has to be written in capitals, or as abbreviation (also in capitals).**

Noted and corrected as suggested.

**Line 629: In Tesche et al. (2009), Müller is not correctly parsed.**

Noted and corrected.

**Line 635: Fernández Rodríguez and Ángela Gonzalo Garijo are not correctly parsed, and no hyphens in the names should be written. (author={Tormo Molina, R. and Maya Manzano, J. M. and Fern\'andez Rodr\'iguez, S. and Gonzalo Garijo, \'A and Silva Palacios, I.})**

Corrected as suggested.

---

## Author Response (AR2)

Dear Editor,

We have added a paragraph in the Abstract and Conclusions related to the correlation of information between the surface and the lidar observations which are at an elevated height. We have also added the limitations of the present study and where future efforts should be focused on.

Kind regards,

Maria